

# High temperature series expansions of S = 1/2 Heisenberg spin models: Algorithm to include the magnetic field with optimized complexity

**Laurent Pierre[1], Bernard Bernu[2★] and Laura Messio[2,3†]**

**1** Université de Paris Nanterre, Nanterre, France
**2** Sorbonne Université, CNRS, Laboratoire de Physique Théorique de la Matière Condensée, LPTMC, F-75005 Paris, France
**3** Institut Universitaire de France (IUF), F-75005 Paris, France

★ bernard.bernu@sorbonne-universite.fr , † laura.messio@sorbonne-universite.fr

## Abstract

This work presents an algorithm for calculating high temperature series expansions (HTSE) of Heisenberg spin models with spin $S = 1/2$ in the thermodynamic limit. This algorithm accounts for the presence of a magnetic field. The paper begins with a comprehensive introduction to HTSE and then focuses on identifying the bottlenecks that limit the computation of higher order coefficients. HTSE calculations involve two key steps: graph enumeration on the lattice and trace calculations for each graph. The introduction of a non-zero magnetic field adds complexity to the expansion because previously irrelevant graphs must now be considered: bridged graphs. We present an efficient method to deduce the contribution of these graphs from the contribution of sub-graphs, that drastically reduces the time of calculation for the last order coefficient (in practice increasing by one the order of the series at almost no cost). Previous articles of the authors have utilized HTSE calculations based on this algorithm, but without providing detailed explanations. The complete algorithm is publicly available, as well as the series on many lattice and for various interactions.

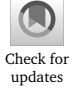

# 1   Introduction

Frustrated quantum spin models exhibit unconventional phases possessing properties as fractional excitations or gapless spin liquid character [1] and are now realized in more and more materials. However, even for spin interactions as simple as Heisenberg, the nature of the ground state is still debated on some non bipartite antiferromagnetic lattices. In these cases, frustration prevents the use of exact methods (exact means here with only statistical errors) such as path integral quantum Monte Carlo or stochastic series expansions [2].

     Studying these models requires the use of specific tools that are in permanent evolution: variational methods [3], mean-field methods [4], tensor-product numerical methods [5, 6], renormalization group methods [7]... A last tool is series expansions, declined in many versions depending on the variable used: the inverse spin length $1/S$ for spin wave theory [8], an interaction strength $\lambda$ for perturbation theory [9, 10], the temperature $T$ for low-$T$ expansions [11] (requiring discrete excitations and the knowledge of the ground state). High temperature series expansions (HTSE) in the inverse temperature $\beta = 1/T$ offer distinct advantages. They are insensitive to frustration, directly address the thermodynamic limit without requiring finite-size scaling and do not require any knowledge on the system (the zeroth order is the infinite temperature limit).

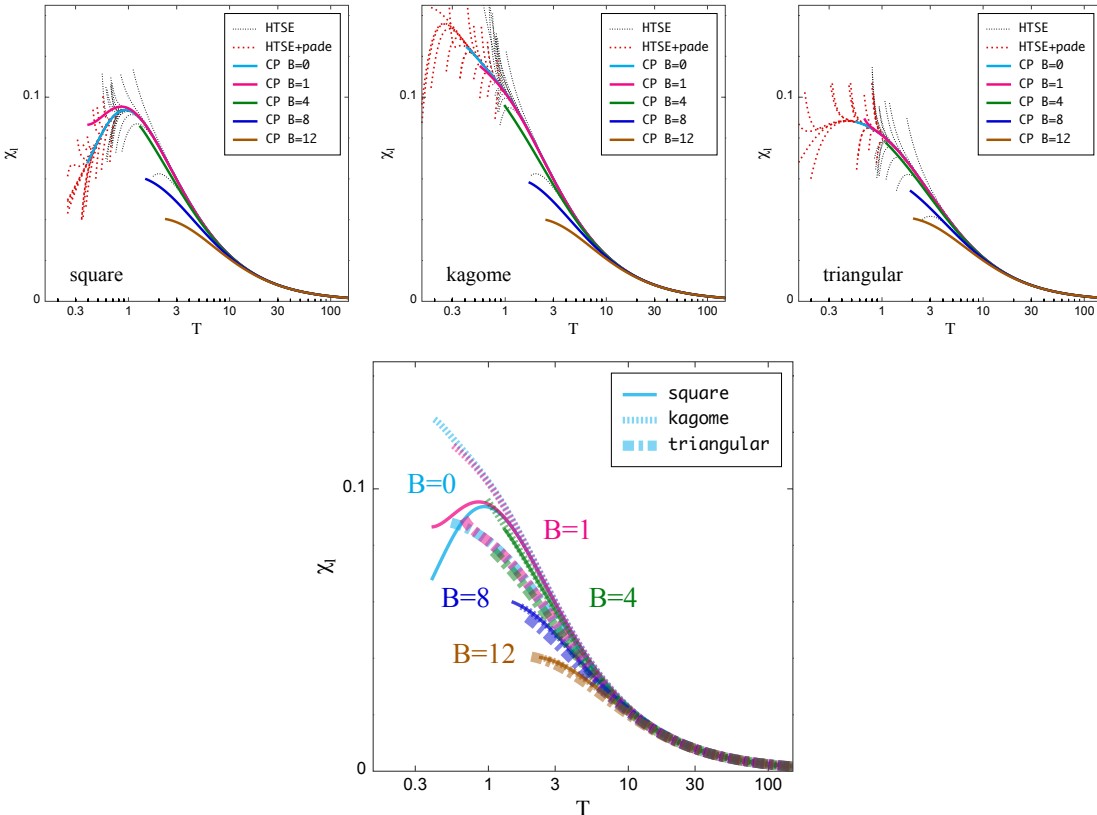

Figure 1: Linear magnetic susceptibility $\chi_l$ at order $n$ in $\beta$, $n = 19$ for square and kagome and $n = 17$ for triangular lattices with antiferromagnetic first neighbor interactions $J = 1$. Top: the raw series at orders 2 to $n$ for $B = 0$ (black dotted lines), the $n + 1$ Padé approximants (PA) of the $B = 0$—series at order $n$ (red dotted lines). These curves are cut when they stray too far from each other. CP curves stand for coinciding PAs, i.e. when at least 8 of them agree (differences less than 0.05). Bottom: comparison of the CP for these three lattices.

The raw HTSE can be used either without any extrapolation method or with Padé approximants [12] to fit the HT measurements of the specific heat $c_V$ and of the linear magnetic susceptibility $\chi_l = m/B$, ($m$ is the magnetization per site and $B$ the magnetic field), or to compare with other theoretical methods. The Curie law [13, 14] is the most simple example of HTSE, where the order 3 allows to fit $\chi_l^{-1}(T)$ at high $T$ with a line crossing the $T$-axis at a so-called Curie temperature, indicative of the energy scale (actually a linear combination of all the exchanges) of the compound.

Determination of high-order series find their roots in the 60's [12, 15, 16]. The explosion of computational power and the development of improved algorithms have allowed to get further orders. HTSE to high orders for $S = \frac{1}{2}$ Heisenberg interactions have been obtained for example on the diamond lattice to order $n = 14$ [10], triangular [17], pyrochlore [18], square, simple cubic and bcc [19] to $n = 13$, fcc [19] to $n = 12$, hyperkagome to $n = 16$ [20], Shastry-Sutherland to $n = 10$ [21], etc. Higher spins, other lattices and interactions have also been considered [10, 11, 22–26]).

But the magnetic field **B** was rarely included, except at first order (giving $\chi_l$ at **B** = 0). However, **B** is an experimentally adjustable parameter known to induce various unexpected phenomena such as magnetization plateaus and phase transitions [27, 28]. Recent advances

have allowed the generation of extreme magnetic fields reaching up to 140T [29], thus expanding the possibilities for material investigation. It is why we are going to pay a special attention to **B** in the following.

HTSE offers valuable insights into the high temperature regime, where temperatures exceed the typical interaction strength. Their finite convergence radius, which is tied to this energy scale, defines a finite temperature $T_r$ below which adding more terms to the series does not improve the results quality when we just sum the first terms of the series. This phenomenon is illustrated on Fig. 1 for the square, kagome and triangular lattices with antiferromagnetic first neighbor Heisenberg interactions $J$, where whatever the lattice, $T_r \simeq J$ (dotted black lines) for $B = 0$. The series of $\chi_l$ has here been calculated for any value of **B** up to order $n = 19$ for the square and kagome lattice , and to $n = 17$ for the triangular lattice (publicly available on [30]). Turning on **B** still reduces the interval of temperature where the series converges (Fig. 1).

Extrapolation techniques have been developed to extend the analysis to lower temperatures. We detail them below to illustrate how the series are used, although we will only treat the calculation of the coefficients. The simplest extrapolation method is the use of Padé approximants: a function $f$ is approximated by a ratio of two polynomials $R^{[p,q]} = P_p/Q_q$ with degrees $p$ and $q$, such that the Taylor expansion of $R^{[p,q]}$ matches the one of $f$ up to order $n = p + q$. Padé approximants coincide (up to an arbitrary definition) down to temperatures typically lower than $T_r$. An illustration of CP (coinciding Padé approximants) is given on Fig. 1, where they allow to go down to $T \simeq 0.3J$ for $B = 0$.

In systems exhibiting a finite temperature phase transition, any extrapolation method is confronted to the free energy singularity at the inverse critical temperature $\beta_c = 1/T_c$, staying on the real axis [31]. However, information on critical exponents and on $T_c$ can be extracted from the coefficients using the Dlog-Padé or ratio method [19,32–34] and be used in extrapolation techniques [35,36].

When no singularity is expected on the real $\beta$ axis (when the system orders at $T = 0$ or does not order at all), the entropy method [24,37–39] incorporates hypotheses on the low energy physics to propose extrapolations down to zero temperature. These hypotheses are the nature of the low energy excitations and the ground state energy $e_0$. When $e_0$ is unknown, it can be determined in a self-consistent manner [24]. In a few words, the entropy method consists in changing the thermodynamic variable $\beta \to e$ and working with the series of the entropy $s(e)$ in the variable energy $e$ (the coefficients of the new series is obtained directly from the HTSE coefficients). An (expected to be) analytic function $G(e)$ is constructed, that depends on $s(e)$, carefully treating the singularity of $s(e)$ at $e = e_0$. $G(e)$ can be safely extrapolated by Pade approximants. Following the reverse path, one goes back to $s(e)$ and gets functions of $\beta = \frac{\partial s}{\partial e}$.

The entropy method has been applied to extrapolate $c_V$ and $\chi_l$ and compare with experimental results on several compounds: vanadium oxyfluoride $(NH_4)_2[C_7H_{14}N][V_7O_6F_{18}]$ [40], herbertsmithite $ZnCu_3(OH)_6Cl_2$ [41, 42] and its polymorph kapellasite [43], $Ba_8CoNb_6O_{24}$ [25]. In all these examples, a model Hamiltonian was proposed, whose parameters were fitted to the measurements. This complements ab initio methods such as DFT [44]. When fitting the model parameters, a common practice is to only vary the temperature $T$, but the constraints could be significantly enhanced by considering the $(T, B)$ plane instead. This highlights the importance of computing HTSE for any **B**.

Whatever the value of **B**, any extrapolation methods requires the largest possible number of series coefficients. We insist on the fact that knowing a series up to some order $n$ means that correlations in any size-$n$ cluster are exactly taken into account. Calculating just one more coefficient is hard, but it brings a strong constraint on extrapolations.

This article introduces an algorithm designed to calculate the series efficiently for a Heisenberg model with $S = \frac{1}{2}$ spins, in the presence of a magnetic field **B**. Sec. 2 is devoted to an extensive presentation of the HTSE method, and of the difficulty to get expansion with $\mathbf{B} \neq 0$ due to the contribution of clusters with bridges. Sec. 3 presents an algorithm to calculate the contribution of these clusters, which is used in Sec. 4 to calculate contribution of trees. Sec. 5 is the discussion and conclusion. Along this article, some proofs have been kept for Appendices C and D, together with a recall of some vocabulary on graphs in A and of cumulant properties in B.

## 2  High temperature series expansions (HTSE) for Heisenberg $S = \frac{1}{2}$ models

We consider a periodic spin lattice (1 dimension: chains, ladders, 2 dimensions: square, triangular, honeycomb, kagome, 3 dimensions: cubic, face centered cubic, pyrochlore...), with short-range interactions (first, second, third... neighbors). HTSE can include spin anisotropies, Dzyaloshinskii-Moriya... , even if only Heisenberg interactions are considered in the following. Multispin interactions (also called ring or cyclic exchange) are possible [45], increasing the complexity of the graph enumeration. In this case, at order $n$ in $\beta$, graphs would not only be constituted of $n$ elementary blocks of site or link type, but also of plaquette type (of typically 4 of 6 links). Any spin length can be chosen for HTSE (classical, or any half-integer quantum value [46]). Here, we focus on $S = \frac{1}{2}$.

For any quantity $A(\beta) = \sum_{k=0}^{\infty} a_k \beta^k$, only truncated HTSE are generally accessible, with a finite number of known coefficients $a_{k \leq n}$ (except when the model is analytically solvable, as for example two-dimensional Ising models without magnetic field). Part of the job is to exploit these coefficients to get the largest amount of information, as detailed in the introduction (extrapolation down to the lowest temperature, determination of the exponents of phase transitions if applicable). Here, we concentrate on the initial step, consisting in getting the largest possible number of coefficients, which itself splits in two sub-steps (detailed below): (i) enumerating simple connected graphs $G$ on the lattice, (ii) calculating their contribution $F(G)$ through trace operators (averages at infinite temperatures).

The computation time depends on the model: lattice geometry, spin length and interaction type. The coordination number of each site (related to the lattice geometry and the type of links: first, second... neighbors) determines the evolution of the graph number with the order, whereas the spin type (quantum, classical) and interactions (Dzyaloshinskii-Moriya, anisotropic, cyclic... ) are related to the complexity to calculate averages (traces) for a given graph.

The system is submitted to a magnetic field $\mathbf{B} = B\mathbf{e}_z$ along an arbitrary direction $z$. We define $h = g\mu_B B$ where $g$ is the g-factor and $\mu_B$ the Bohr magneton. For a Hamiltonian that preserves the total spin, the number of (connected) graphs contributing to the HTSE considerably increases when $B$ is switched on, thus reducing the reachable expansion order. Concretely, graphs with bridges or leaves (see Fig. 2 and next section for definitions) are the majority. At order equal to their number of links, they do not contribute when $h = 0$. For $h \neq 0$, they do. In this article, we present an algorithm that reduces the complexity of trace calculations on these graphs in the case of a quantum $S = \frac{1}{2}$ Heisenberg model: in practice, it allows the calculation of one supplementary order as compared with the naive algorithm (note that each additional order needs an order of magnitude more computational time).

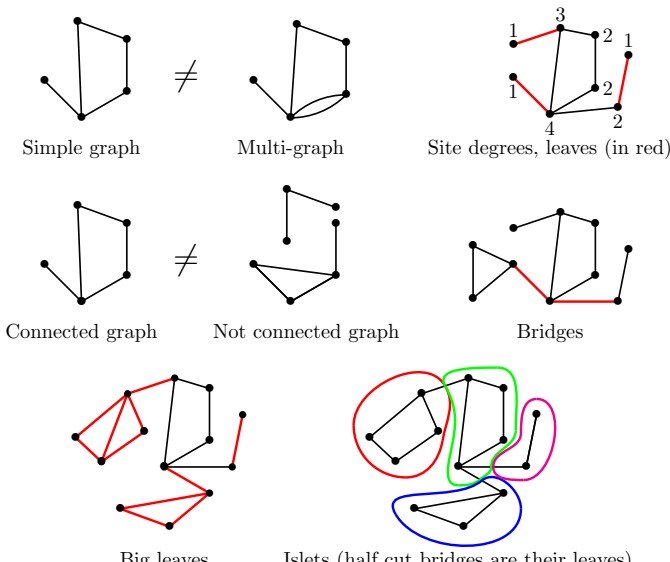

Figure 2: Illustration of graph properties (see App. A for the detailed definitions). The degree $d°s$ of a site $s$ is the number of links emanating from it.

In the first subsection (Sec. 2.1), we define the model and explain how to get series expansion on a finite cluster. In the next one (Sec. 2.2), we switch to the thermodynamic limit, using contributions of finite graphs. Sec. 2.3 shows how to calculate the HTSE coefficients with integers. Finally, in Sec. 2.4 and 2.5, we discuss the complexity of the two main steps of the expansion (graph enumeration and trace calculation) and explain why bridged graphs, and among them, trees, have the largest contribution to the trace computation time.

## 2.1 Definitions

As a first step, we consider a simple connected sub-graph $G$ of the infinite lattice, with $N_s$ sites and $N_l = \#G$ links ($\#G$ is the cardinal of $G$, since for us a graph is a set of links). Fig. 2 illustrates the notion of connected and simple graphs, as well as other graph properties used in the following. The Hamiltonian $H$ of the Heisenberg model on graph $G$ splits into a sum $H_J$ on links $l = l_1 \leftrightarrow l_2$, and a sum $H_B$ on sites:

$$H = -\underbrace{\sum_{l \in G} 2 J_l \, \mathbf{S}_{l_1} \cdot \mathbf{S}_{l_2}}_{H_J} - \underbrace{h \sum_{i, \text{ site of } G} S_i^z}_{H_B} \, . \tag{1}$$

$J_l$ gives the strength of the Heisenberg interaction of link $l$. Note the conventional choice of a positive $J_l$ for ferromagnetic interactions. From now on, we only consider quantum $S = 1/2$ spins, and the scalar product of the spin operator vectors can thus be expressed in terms of permutation operators:

$$\mathbf{S}_{l_1} \cdot \mathbf{S}_{l_2} = \frac{P_l}{2} - \frac{1}{4} \, , \tag{2}$$

where $P_l$ exchanges the spin states on the two sites of link $l$. Up to an unimportant additive constant $J_l/2$ for each link term of $H_J$, the exchange Hamiltonian on $G$ now reads:

$$H_J = -\sum_{l \in G} J_l P_l \, . \tag{3}$$

We are interested in the infinite lattice properties, but as an intermediate step, we calculate the logarithm of the partition function $Z$ on $G$, that we expand in $\beta$ (this is called the fixed-$B$ expansion):

$$\ln Z(\beta) = \ln\left(\mathrm{Tr}\, e^{-\beta H}\right) = \ln\left(\frac{\mathrm{Tr}\, e^{-\beta H}}{\mathrm{Tr}\,\mathbf{I}} 2^{N_s}\right)$$

$$= \ln(2^{N_s}) + \ln\left(1 + \sum_{n=1}^{\infty} \frac{\langle (-\beta H)^n \rangle}{n!}\right) \tag{4}$$

$$= N_s \ln 2 + \sum_{n=1}^{\infty} \frac{[(-\beta H)^{(n)}]}{n!}. \tag{5}$$

The trace, $\mathrm{Tr}$, is taken over states of an orthonormal basis $\mathcal{B} = \{|\phi_i\rangle, i = 1\ldots 2^{N_s}\}$ of the spin configurations. $\mathbf{I}$ is the identity operator. The averages $\langle . \rangle$ of Eq. (4) are defined as $\langle A \rangle = \frac{\mathrm{Tr}\, A}{\mathrm{Tr}\,\mathbf{I}} = \frac{1}{2^{N_s}}\sum_{i=1}^{2^{N_s}} \langle \phi_i | A | \phi_i \rangle$. The cumulant of order $n$ of $-\beta H$ is denoted $[(-\beta H)^{(n)}]$ or $[-\beta H, -\beta H, \ldots, -\beta H]$ to be distinguished from $[(-\beta H)^n]$ which is a first order cumulant equal to $\langle (-\beta H)^n \rangle$, the moment of order $n$ of $-\beta H$. App. B recalls definitions and some relations between averages, moments and cumulants. Expanding $(-\beta H)^n$ using Eq. (3) gives a sum of terms, each of them corresponding to a list of $n_l$ undirected links and $n_s$ sites of $G$, with $n_l + n_s = n$.

The aforementioned expansion combines both links and sites. But we now use an expansion solely involving clusters of links and exactly evaluate the contribution of $H_B$ at each order in $H_J$. From a thermodynamic standpoint, this corresponds to a transformation of the ensemble $(\beta, h)$ to $(\beta, \beta h)$, where $\beta h$ is a new thermodynamic variable fixed in the $\beta$-expansion [47].

From now on, we write $J$ instead of $\beta J$ and $h$ instead of $\beta h$ to lighten notations. $\beta$ can be reinjected in the formulae using the inverse transformation when needed.

We denote $S^z = \sum_i S_i^z$ the total magnetization along the $z$ direction. Additionally, we define several variables associated to $h$ for future use:

$$Y = e^{\frac{h}{2}} + e^{-\frac{h}{2}} = 2\cosh\frac{h}{2}, \qquad \theta = \theta_+ - \theta_- = \tanh\frac{h}{2}, \tag{6}$$

$$\theta_+ = \frac{1+\theta}{2} = \frac{e^{h/2}}{Y}, \qquad\qquad \theta_- = \frac{1-\theta}{2} = \frac{e^{-h/2}}{Y}. \tag{7}$$

Averages and cumulants are now taken with respect to a different measure (proportional to $e^{hS_z}$ for each element of a basis of $S^z$-eigenvectors). This alternative expansion of $\ln Z(\beta)$ in powers of $\beta$ will be referred to as the fixed-$\theta$ expansion:

$$\ln Z(\beta) = \ln\left(\mathrm{Tr}\, e^{-\beta H}\right) = \ln\left(\frac{\mathrm{Tr}\left(e^{-\beta H_J} e^{hS^z}\right)}{\mathrm{Tr}\left(e^{hS^z}\right)} Y^{N_s}\right)$$

$$= \ln\left(Y^{N_s}\right) + \ln\left(1 + \sum_{n=1}^{\infty} \frac{\langle\langle (-\beta H_J)^n \rangle\rangle}{n!}\right) \tag{8}$$

$$= N_s \ln Y + \sum_{n=1}^{\infty} \frac{[\![(-\beta H_J)^{(n)}]\!]}{n!}, \tag{9}$$

where $\mathrm{Tr}\left(e^{hS^z}\right) = Y^{N_s}$. The obtained formulae are similar to those of the uniform measure (4) and (5). With this non-uniform measure, the average of an operator $A$ is denoted: $\langle\langle A \rangle\rangle = \frac{\mathrm{Tr}(Ae^{hS^z})}{\mathrm{Tr}(e^{hS^z})}$. The moment and cumulant of a multiset (or list) $L$ of operators commuting with $S^z$ are denoted $\langle\langle L \rangle\rangle$ and $[\![L]\!]$. Only lists of $n$ links now appear in the term of order $n$ of the $\beta$-expansion (such expansions were previously derived in [48] and discussed but not used in [16]).

We define $g$ and $\overline{g}$, and their expansions in powers of $H_J$ or in powers of $J$ as

$$\overline{g}(G) = \frac{\text{Tr}\left(e^{-\beta H_J} e^{hS^z}\right)}{Y^{N_s}} = \sum_{n=0}^{\infty} \frac{\langle\langle (-\beta H_J)^n \rangle\rangle}{n!} = \sum_{U \in \mathbb{N}^G} \frac{J^U}{U!} \langle\langle U \rangle\rangle, \tag{10}$$

$$g(G) = \ln \overline{g}(G) = \sum_{n=1}^{\infty} \frac{[\![ (-\beta H_J)^{(n)} ]\!]}{n!} = \sum_{U \in \mathbb{N}^G} \frac{J^U}{U!} [\![ U ]\!]. \tag{11}$$

$U \in \mathbb{N}^G$ is a mapping of $G$ into $\mathbb{N}$. Hence $U$ is also a multigraph whose support is a part of $G$ (or a multiset of elements of $G$) in which a link $l$ has a multiplicity $U(l)$. Numerator $J^U$ is $\prod_{l \in G} J_l^{U(l)}$ (we recall that its order in $\beta$ is $\#U = \sum_l U(l)$). Denominator $U!$ is $\prod_{l \in G} U(l)!$. In $\langle\langle U \rangle\rangle$ and $[\![ U ]\!]$ a link $l \in U$ is identified to $P_l$. Note that $\langle\langle \emptyset \rangle\rangle = 1$ and $[\![ \emptyset ]\!] = 0$, and for a single link $l$:

$$\langle\langle l \rangle\rangle = [\![ l ]\!] = \frac{\text{Tr}\left(P_l e^{hS_z}\right)}{Y^2} = \theta_+^2 + \theta_-^2 = \frac{1 + \theta^2}{2}. \tag{12}$$

More generally, for any multigraph $U$, the moment $\langle\langle U \rangle\rangle$ and cumulant $[\![ U ]\!]$ are even polynomials in $\theta$ the degrees of which verify $d_\theta^\circ \langle\langle U \rangle\rangle \leq N_s(U)$ and $d_\theta^\circ [\![ U ]\!] \leq 2\#U$ (proof in App. D.1). The average of the product of independent variables is the product of their averages. Hence for a not connected multigraph $U$ with connected components labelled $U_1, U_2...$, we have $\langle\langle U \rangle\rangle = \prod_i \langle\langle U_i \rangle\rangle$ and $[\![ U ]\!] = 0$ (proof in App. B.5).

$\langle\langle U \rangle\rangle$ and $[\![ U ]\!]$ are in fact independent of the graph $G$ for any multi-graph $U$ of the infinite lattice: they are the same for two different simple graphs $G_1$ and $G_2$ including the support of $U$. Thus they are well defined in the thermodynamic limit, and can be evaluated on the smallest possible graph $G$: the support of $U$.

## 2.2 From a finite graph to the infinite lattice

We now discuss the thermodynamic limit, by first taking a finite periodic lattice $\mathcal{L}$ of $N_{uc}$ unit cells, each containing one or several sites. The series expansions described in the previous subsection are valid on $\mathcal{L}$, and each term of order $n$ of Eq. (9) is a sum over connected multigraphs $U$ of $\mathcal{L}$ with $n$ links. A multi-graph $U$ without topologically non trivial loops is by definition equivalent to $N_{uc}$ graphs up to a translation on $\mathcal{L}$. If $n_m$ is the minimal number of links of a topologically non trivial loop on $\mathcal{L}$, we can group multi-graphs into equivalence classes of $N_{uc}$ elements up to order $n_m - 1$. The HTSE of $\ln Z(\beta)/N_{uc}$ truncated at some order $n$ thus does no more depend on the lattice size when $\mathcal{L}$ is large enough: it results in a well defined HTSE in the thermodynamic limit.

To determine this expansion, we list translation-equivalent-classes of connected simple graphs $G$ on the infinite lattice. For a representative of each class, we then determine $F(G)$, the sum of the contributions of all multi-graphs $U$ whose support is exactly $G$:

$$F(G) = \sum_{U \in \mathbb{N}_{>0}^G} \frac{J^U [\![ U ]\!]}{U!}, \tag{13}$$

where $\mathbb{N}_{>0}$ is the set of positive integers. The classes of translation-equivalent graphs can still be regrouped in larger classes of topologically equivalent (isomorphic) graphs $\overline{G}$, carefully keeping track of the weak embedding constant of each class $w(\overline{G})$ (in other words, the occurrence number per unit cell).

For models with several types of links, graph isomorphisms must preserve $J_l$ (type of link) in order to ensure that $J^U$ and $F(G)$ are well defined in Eq. (13). In other words, when two simple graphs $\overline{G}$ and $\overline{G'}$ only differ by their $J_l$, $F(\overline{G})$ and $F(\overline{G'})$ are not simply related. But $F(G) = \frac{J^G[\![G]\!]}{G!} + o(\beta^{N_l})$, hence $F(\overline{G'}) = \frac{J^{\overline{G'}}}{J^{\overline{G}}}F(\overline{G}) + o(\beta^{N_l})$ may be used when $n = N_l$.

To simplify the notations in this article, only one type of $J$ is used in the following. Anyway we need $w(\overline{G})$ and $F(\overline{G})$ for each class $\overline{G}$, that we inject in the so-called *linked-cluster expansion* of $g(\mathcal{L})$ in the thermodynamic limit:

$$g_\infty = \sum_{\overline{G}} w(\overline{G})F(\overline{G}). \tag{14}$$

$F(G)$ can be deduced from the *inclusion-exclusion formula*, valid in any linked cluster expansion (deduced from Eqs. (11) and (13)):

$$F(G) = g(G) - \sum_{G' \subsetneq G} F(G'). \tag{15}$$

Such linked cluster expansions are used in various contexts, for example, in the numerical linked cluster expansions [49–51], where the $F(G)$ are calculated exactly for all $G$-classes up to some cluster size and the free energy is calculated via a truncation in the cluster size, or more recently in a projective cluster-additive transformation [52], where the low energy sectors of a perturbed Hamiltonian are explored. In HTSE, the $F(G)$ expansion is truncated at order $n$ in $\beta$.

Note that $F(G)$ only contributes at orders $n \geq \#G$. Thus, to get the HTSE up to some order $n$, we need to enumerate all simple connected graphs $G$ with $\#G \leq n$ (Sec. 2.4), and for each of them, calculate $F(G)$ up to order $n$ (Sec. 2.5), trying in each step to identify the most time consuming step and to optimize it.

## 2.3 Integerness during calculation and storage of results

$\overline{g}$, $g$ and $F(G)$ are polynomials in the variables $J_l$, where the coefficients are themselves polynomials in $\theta^2$, where the coefficients are rational numbers. Here, we first justify the use of the common denominator $2^k k!$ for these rational numbers, with $k$ their order in $J$. We give examples of how it allows to only store integer numerators with implicit denominator during computations. We finally give the definition of the series coefficients that we have used in our code [53].

*Proof.* Let $G$ be the graph of links of (partial) Hamiltonian $H_J$. When expanded, $H_J^k$ is a weighted sum of permutations of the $N_s$ sites of $G$ (see Eq. 3). Such a permutation $\sigma$ writes as a product of disjoint cycles of lengths $r_1, r_2, \ldots$. Let $r = \sum_i (r_i - 1)$. Then $r \leq k$. Furthermore $\sigma$ has after division by $Y^{N_s}$ a trace $\langle\!\langle \sigma \rangle\!\rangle = \prod_i (\theta_+^{r_i} + \theta_-^{r_i})$ (as only configurations with all spins up or down on each cycle are unchanged by $\sigma$). But $\theta_+^{r_i} = (\frac{1+\theta}{2})^{r_i} \in \mathbb{Z}_{r_i}[\theta]/2^{r_i}$, where $\mathbb{Z}_{r_i}[\theta]$ denotes the set of integer polynomials of degree at most $r_i$ in variable $\theta$. Furthermore as $\theta_+^{r_i} + \theta_-^{r_i}$ is twice the even part of $\theta_+^{r_i}$ in $\theta$, it is in $Z_{\lfloor r_i/2 \rfloor}[\theta^2]/2^{r_i-1}$ (the floor function $\lfloor x \rfloor$ or $x$ is the greatest integer less than or equal to $x$). But $\lfloor r_i/2 \rfloor \leq r_i - 1$, hence $\langle\!\langle \sigma \rangle\!\rangle \in \mathbb{Z}_r[\theta^2]/2^r$. Therefore $\langle\!\langle H_J^k \rangle\!\rangle \in (\mathbb{Z}_k[\theta^2])[J]/2^k$, as well as $[\![H_J^{(k)}]\!]$, since cumulants are homogeneous integer polynomials of moments according to Eq. (B.9). We can use denominator $2^k k!$ in terms of order $k$ in $J$ within $\overline{g}(G) = \sum_k \langle\!\langle H_J^k \rangle\!\rangle/k!$, $g(G) = \sum_k [\![H_J^{(k)}]\!]/k!$ and $F(G) = g(G) - \sum_{G' \subsetneq G} F(G')$, and even within $(1 - \overline{g}(G))^i/i$ despite division by $i$, because it is the sum of all products of $i$ moments in Eq. (B.9). This last term is part of the expansion of $\ln \overline{g}(G) = g(G)$ which we use to calculate $g(G)$. $\qquad\square$

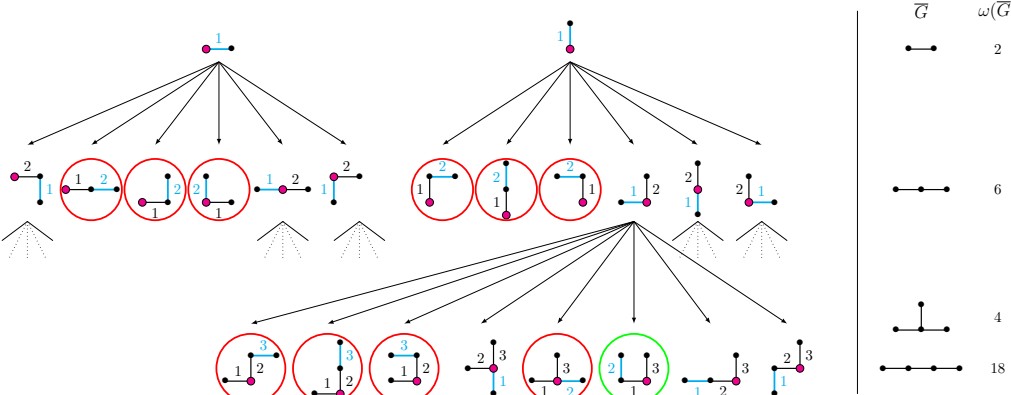

Figure 3: Enumeration of graphs on a lattice: example of the square lattice. We depart from the first generation: each link in the unit cell, the root (empty graph) is not represented. In the red-circled graphs, the newly added link (in cyan) has not the smallest possible label. Such a graph has no children and only its double with the smallest label, child of another parent, is allowed to breed. The green circle highlights a graph, which is kept, although the label of its new link is 2 instead of 1. But the link of label 1 is a bridge, which cannot have been added to a parent (an orphan link). On the right are recapitulated the topological graphs and their occurrence numbers.

**Remark 1:** During the calculation of the product $a\frac{J^i}{i!2^i} \times b\frac{J^j}{j!2^j} = ab\binom{i+j}{i}\frac{J^{i+j}}{(i+j)!2^{i+j}}$ of two terms of orders $i$ and $j$, we replace the multiplication of the two rational numbers $a/i!2^i$ and $b/j!2^j$ by a multiplication of the three integers $a$, $b$ and $\binom{i+j}{i}$.

**Definition of the series coefficients using implicit denominators:** To store the series in a uniform way, we define the coefficients of a HTSE by:

$$\frac{\ln Z(\beta, \theta)}{N_s} = \ln Y + \frac{1}{n_{\text{uc}}} \sum_{k=1}^{n} \frac{1}{2^k k!} \sum_{r=0}^{k} D_{k,r} \theta^{2r} + O(\beta^{n+1}), \tag{16}$$

with $n_{\text{uc}}$ the number of sites in a unit cell and $D_{k,r}$ are homogeneous polynomials of degree $k$ with integer coefficients in the Hamiltonian parameters $J_1$, $J_2$... appearing in Eq. 1 and implicitly multiplied by $\beta$. In practice, the files generated by our code [53] store the coefficients $D_{k,r}$. They are publicly available [30].

## 2.4 Enumeration of simple connected graphs on a periodic lattice

This part of the calculation consists in finding all relevant simple connected graphs $G$ (those appearing on the considered lattice) and calculating their weight $w(G)$. This is not the main subject of this article, but for completeness, in this section, we present an algorithm that does the job and has the advantage of being parallelizable, as well as two ways of sparing time in some specific situations. It is mathematically described in [54]. A directed tree is constructed, whose vertices are graphs on the lattice (in fact, classes of translation-equivalent graphs). Graphs of the $n$'th generation have $n$ links, and each branch of the tree can be explored independently, as we are able to decide if we keep a vertex or not without exploring the tree (see Fig. 3). The root of the tree is the empty graph. The first generation vertices are all the one-link graphs contained in a unit cell (translationally inequivalent). The next generations are constructed as follows:

- For a graph $G$ with $n-1$ links embedded on the lattice, we consider all the simple graphs with $n$ links obtained by adding an adjacent link to $G$.

- We want to keep only one among all identical (up to a translation) graphs obtained from all $G$'s. For this, the $n$ links of each child $G'$ are labelled in a way that only depends on $G'$ and not of its parent (ordering the coordinates of its sites for example). Thus, for each copy, the label of the new link is different. Note that bridges of $G'$ are orphan links, meaning that they and they alone cannot be new links. We keep $G'$ only if the new link has the smallest label among the non-orphan links.

- For each graph $G$ (each vertex of the tree), a canonical label $\overline{G}$ is calculated, that is the same for isomorphic graphs (i.e. identical up to a vertex renumbering). Canonical labels can be calculated using McKay's algorithm [55,56]. All graph isomorphism classes are collected and their occurrence number $w(\overline{G})$ (also called the lattice constant, or weak embedding constant) is counted.

Note that different methods, said more efficient but not implemented in our code, are described in the literature [11,57]. They consist roughly in a first step generating all topological classes of graphs (this step itself can be realized in different ways), and in a second step counting their embedding number on the lattice. It avoids the costly step of the canonical label calculation, that is however reduced in our algorithm using the two following tricks.

### 2.4.1 Avoid the canonical labelling of graphs with leaves

The calculation of canonical labels in the last step of the graph enumeration is expensive. When all sites of the lattice have the same number of neighbors $z$, we can spare time by avoiding to calculate it for graphs with leaves (see App. A for the definition of a leaf), as the multiplicity of their topological graph can be deduced as follows. Let $\overline{G}$ be a topological connected simple graph containing a leaf $l = u \leftrightarrow v$ with $d°u > d°v = 1$. Let $n_a(\overline{G})$ be the number of automorphisms of $\overline{G}$, i.e. the number of permutations of sites of $\overline{G}$, which map links on links. This number is a by-product of McKay's algorithm. Let $n_e(\overline{G}) = n_a(\overline{G})w(\overline{G})$. This is the number of embeddings (injective mappings of sites and links) of $\overline{G}$ into the lattice (per unit cell). In other words $w(\overline{G})$ counts subgraphs of lattice isomorphic to $\overline{G}$, whereas $n_e(\overline{G})$ counts isomorphisms between $\overline{G}$ and subgraphs of the lattice. $w(\overline{G})$ is deduced from:

$$n_e(\overline{G}) = (z - d°u + 1) n_e(\overline{G} \setminus l) - \sum_{\substack{s \text{ site of } \overline{G} \\ s \neq u, u \leftrightarrow s \notin \overline{G}}} n_e(\overline{G} \cup \{u \leftrightarrow s\} \setminus l), \quad (17)$$

requiring only the calculation of $n_a$ for the graphs appearing in the formula. The needed $w$ are known if we calculate $w(\overline{G})$ in ascending order of $N_s(\overline{G})$.

**Example:** We apply formula (17) to calculate $w(\,\bullet\!-\!\bullet\!-\!\bullet\!-\!\bullet\,)$ on a triangular lattice with $z = 6$. The automorphism numbers $n_a$ of $\bullet\!-\!\bullet\!-\!\bullet\!-\!\bullet$, $\bullet\!-\!\bullet\!-\!\bullet$ and $\triangle$ are pictorially represented on Fig. 4, together with the embedding numbers $w$ of $\bullet\!-\!\bullet\!-\!\bullet$ and $\triangle$. Labelling $s$, $t$, $u$ and $v$ the four sites of $\bullet\!-\!\bullet\!-\!\bullet\!-\!\bullet$, $l = u \leftrightarrow v$ is a leaf with $d°u = 2 > d°v = 1$. We get:

$$n_e(\,\bullet\!-\!\bullet\!-\!\bullet\!-\!\bullet\,) = (z - d°u + 1)n_e(\,\bullet\!-\!\bullet\!-\!\bullet\,) - n_e(\,\triangle\,) \quad (18)$$

$$= (6 - 2 + 1)n_a(\,\bullet\!-\!\bullet\!-\!\bullet\,)w(\,\bullet\!-\!\bullet\!-\!\bullet\,) - n_a(\,\triangle\,)w(\,\triangle\,)$$

$$= 5 \times 2 \times 15 - 6 \times 2 = 138.$$

From $n_e(\,\bullet\!-\!\bullet\!-\!\bullet\!-\!\bullet\,) = n_a(\,\bullet\!-\!\bullet\!-\!\bullet\!-\!\bullet\,)w(\,\bullet\!-\!\bullet\!-\!\bullet\!-\!\bullet\,)$, we deduce $w(\,\bullet\!-\!\bullet\!-\!\bullet\!-\!\bullet\,) = 69$.

Figure 4: Quantities used in the calculation of $w(\,\bullet\!-\!\bullet\!-\!\bullet\!-\!\bullet\,)$ on the triangular lattice, Eq. (18).

**Remark:** The time saved this way is important, as graphs with leaves are the majority when the number of links and the lattice dimensionality increases. In the case of a $d-$dimensional hypercubic lattice [58], $w(\overline{G}) = O((2d-1)^{N_l})$ for a tree of $N_l$ bonds in the limit of large $d$, whereas a topological graph with a loop of $2s$ sites has $w(\overline{G}) = O((2d-1)^{N_l-s})$.

When adding a link to a connected graph, no more than two leaves may disappear. Hence we can prune a graph $G$ with more than $2(n - \#G)$ leaves.

### 2.4.2 Expansion in the magnetic field $B$: Non-contributing graphs

We have seen in Sec. 2.1 an elegant way to get HTSEs which include all orders in the magnetic field $B$, through expansion coefficients that are even polynomials in $\theta$ (fixed-$\theta$ expansion). However, most physical studies are performed at fixed $B$, requiring either to expand the fixed-$\theta$ expansion coefficients of Eq. (9) in powers of $\beta$, or to directly work with the fixed-$B$ expansion of Eq. (5). The final coefficients are of course the same in both cases, and the coefficients in $\beta^l$ are even polynomials in $B$ of maximal order $l$.

To get the fixed-$B$ expansion of $F(G)$ for a graph $G$ up to order $\beta^n$ from the fixed-$\theta$ expansion, the polynomial coefficient $P_l(\theta)$ of the $\beta^l$ term of the latter can be truncated at order $k = n-l$ in $\theta$, but it generally does not bring a lot, except in some cases where $P_l(\theta)$ is divisible by $\theta^{k+1}$. Then, the graph $G$ can simply be discarded. Here are some simple situations where it occurs:

1. For $B = 0$, a graph $G$ with $N_{bf}$ links that are either bridges or leaves can be discarded if $\#G + N_{bf} > n$.

2. A graph $G$ with $N_L$ big leaves (see App. A) does not contribute to the fixed-$B$ expansion at order $n$ if $\#G + N_L > n$.

The proofs are in App. C (they use some formulae derived in the following sections), together with other, better criterion.

### 2.5 Complexity and bottleneck of HTSE

We now evaluate the complexity of calculating $F(G)$ up to order $n$. In the following $J_l$'s may all have the same value, or several values (for example first and second neighbor interactions). For instance a polynomial of degree $n$ in $\theta, J_1, J_2, \ldots, J_k$ has $O(n^{1+k})$ coefficients. Multiplication of two such polynomials takes time $O(n^{2+2k})$. But for simplicity, time complexity estimates here assume that the $J_l$'s are all equal and this time is $O(n^4)$. The calculation of $F(G)$ divides in three successive steps, whose complexities are now given (proofs in App. D):

- Get the averages $\langle\!\langle H_J^k \rangle\!\rangle$ for $k \leq n$, in a time $O(4^{N_s} n N_l/\sqrt{N_s})$. According to Eq. (10) we have $\overline{g}(G)$ at order $n$.

- Calculate $g(G)$ as $\ln \overline{g}(G)$ at order $n$ in a time $O(n^4 N_s)$.

- From $g(G)$ and $F(G')$ for $G' \subsetneq G$, calculate $F(G)$ using Eq. (15) in a time $O(2^{N_l} n^2)$, or better in a time $O(N_l^2 n^2)$ as explained in App. D.1.

Finally, the bottleneck to get $F(G)$ at order $n$ among the three steps listed above is the calculation of averages in $O(4^{N_s} n N_l / \sqrt{N_s})$. Then, at fixed $n$, the most greedy graphs are those with the largest $N_s$. As the considered graphs are connected, we have the condition $N_s \leq 1 + N_l \leq n + 1$. For $n$ fixed, the way to maximize $N_s$ is to choose $N_l = n$ and to forbid loops ($N_s = 1 + N_l$), which results in graphs that are trees with $n$ links and a complexity in $O(4^n n^{3/2})$.

The next section describes a way to calculate $F(G)$ in a considerably faster time $O(n^2)$, for bridged graphs with $N_l = n$ links (which include all trees except the star graph $T_n$ of Fig. 5, left), assuming that we know $F(G')$ for any simple graph $G' \subsetneq G$.

## 3 $O(n^2)$ complexity for $n$ links bridged graphs and order $n$ expansion

Let $G$ be a simple connected graph with $N_l = \#G$ links. According to Eq. (13), $F(G) = J^G [\![G]\!] + o(J^G)$ and cumulant $[\![G]\!]$ is derived from moments of subgraphs of $G$ by

$$[\![G]\!] = \sum_{q \in \mathcal{Q}(G)} g_0(\#q) \prod_{G' \in q} \langle\!\langle G' \rangle\!\rangle, \tag{19}$$

$$g_0(i) = (-1)^{i-1}(i-1)! = (-1)(-2)\cdots(1-i), \tag{20}$$

where $\mathcal{Q}(G)$ is the set of partitions of $G$ and $\#q$ is the cardinal of the partition $q$. This equation is proved in appendix (B) as Eq. (B.13).

In this section, we demonstrate that if $G$ is a bridged graph (an undirected graph that can be split in two connected components by removing a single link), $F(G)$ can be calculated at order $n = \#G$ in $J$, in time $O(n^2)$, if we know $F(G')$ for any connected subgraph $G' \subsetneq G$.

We choose a bridge of $G$ that we denote $u \leftrightarrow v$. Let $U$ and $V$ be the two connected components of $G \setminus \{u \leftrightarrow v\}$. We assume that $u$ is a site of $U$ and $v$ is a site of $V$.

The first main result of this article is:

$$[\![G]\!] = [\![U, u \leftrightarrow v, V]\!] = \frac{2}{\theta^2} [\![U, u \leftrightarrow v]\!] [\![u \leftrightarrow v, V]\!]. \tag{21}$$

This equation can be pictorially represented on an example as:

$$\left[\!\left[ \begin{array}{c} \end{array} \right]\!\right] = \frac{2}{\theta^2} \left[\!\left[ \begin{array}{c} \end{array} \right]\!\right] \left[\!\left[ \begin{array}{c} \end{array} \right]\!\right] .$$

*Proof.* Operator $P_{u \leftrightarrow v}$ exchanges spins of sites $u$ and $v$ of link $u \leftrightarrow v$. So

$$P_{u \leftrightarrow v} = p_u^{++} p_v^{++} + p_u^{+-} p_v^{-+} + p_u^{-+} p_v^{+-} + p_u^{--} p_v^{--}, \tag{22}$$

where operator $p_s^{\epsilon \epsilon'}$ transforms state $\epsilon$ of spin operator $S_s^z$ into state $\epsilon'$. We define

$$[\![U]\!]_u^+ = [\![U, p_u^{++}]\!], \qquad [\![U]\!]_u^- = [\![U, p_u^{--}]\!]. \tag{23}$$

The trace of an operator which decreases total spin $S^z$ on the sites of $U$, is zero. Hence $\langle\!\langle G', p_u^{+-} p_v^{-+}\rangle\!\rangle = 0$ for any subgraph $G' \subset U \cup V$. Hence $[\![U, p_u^{+-} p_v^{-+}, V]\!] = 0$. When computing moment or cumulant of a graph $G$ with a leaf ($U$ or $V$ being empty) or bridge $u \leftrightarrow v$ we can replace $P_{u \leftrightarrow v}$ by $p_u^{++} p_v^{++} + p_u^{--} p_v^{--}$. Sum of both projections on the possible states of a spin is identity, which is independent of any operator. Hence if $U$ is a non-empty graph: $[\![U]\!]_u^+ + [\![U]\!]_u^- = [\![U, p_u^{++}]\!] + [\![U, p_u^{--}]\!] = [\![U, p_u^{++} + p_u^{--}]\!] = [\![U, \mathbf{I}]\!] = 0$. For an empty graph: $[\![\emptyset]\!]_u^+ = [\![p_u^{++}]\!] = \langle\!\langle p_u^{++}\rangle\!\rangle = \theta_+$ i.e. the probability for an isolated spin to be in the + state.

$$U \neq \emptyset \quad \Rightarrow \quad [\![U]\!]_u^- = -[\![U]\!]_u^+\,, \qquad [\![\emptyset]\!]_u^+ = \theta_+, \qquad [\![\emptyset]\!]_u^- = \theta_-\,. \tag{24}$$

Links in $U$ and $p_u^{\epsilon\epsilon}$ operate on spins of sites of $U$. These operators commute with those of $V$. Using the properties of cumulants of independent sets of operators (see Eq. (B.19)) and the linearity of cumulants, we have:

$$\begin{aligned}
[\![U, u \leftrightarrow v, V]\!] &= [\![U, \sum_{\epsilon \in \{+, -\}} p_u^{\epsilon\epsilon} p_v^{\epsilon\epsilon}, V]\!] \\
&= \sum_{\epsilon \in \{+, -\}} [\![U, p_u^{\epsilon\epsilon}]\!] [\![p_v^{\epsilon\epsilon}, V]\!] \\
&= [\![U]\!]_u^+ [\![V]\!]_v^+ + [\![U]\!]_u^- [\![V]\!]_v^-\,.
\end{aligned} \tag{25}$$

With an empty $V$ (or $U$), this equation becomes:

$$[\![U, u \leftrightarrow v]\!] = [\![U]\!]_u^+ \theta_+ + [\![U]\!]_u^- \theta_- = [\![U]\!]_u^+ (\theta_+ - \theta_-) = [\![U]\!]_u^+ \theta\,, \tag{26}$$

$$[\![u \leftrightarrow v, V]\!] = [\![V]\!]_v^+ \theta\,. \tag{27}$$

Otherwise:

$$[\![U, u \leftrightarrow v, V]\!] = [\![U]\!]_u^+ [\![V]\!]_v^+ + (-[\![U]\!]_u^+)(-[\![V]\!]_v^+) = 2[\![U]\!]_u^+ [\![V]\!]_v^+\,. \tag{28}$$

Combining Eqs. (26), (27) and (28) gives Eq. (21). $\qquad\square$

**Complexity:** Search for bridge $u \leftrightarrow v$ and subgraphs $U$ and $V$ in graph $G$ takes time $O(n)$. Retrieval of $[\![U, u \leftrightarrow v]\!]$ as coefficient of $J^U J_{u \leftrightarrow v}$ in $F(U \cup \{u \leftrightarrow v\})$ takes time $O(n)$, since $[\![U, u \leftrightarrow v]\!] \in \mathbb{Q}_{1+N_l(U)}[\theta^2]$. Multiplication of polynomials $[\![U, u \leftrightarrow v]\!]$ and $[\![u \leftrightarrow v, V]\!]/\theta^2$ takes time $O(n^2)$. So the overall time to compute $[\![G]\!]$ is $O(n^2)$.

## 4 Trees with $n$ links

We show in this section the second main result of this article: for a tree $T$ with $N_l \geq 2$:

$$[\![T]\!] = \frac{1}{2} \prod_{s \in \text{ sites of } T} 2C_{d°s}\,, \tag{29}$$

where $d°s$ is the number of links departing from site $s$, and $C_k$ is recursively defined by:

$$C_1 = \frac{\theta}{2}, \qquad C_{k+1} = \frac{dC_k}{dh} = \frac{1 - \theta^2}{2} \frac{dC_k}{d\theta}\,. \tag{30}$$

*Proof.* Value of $C_1$ is given by equations (21) and (29): When joining trees $U$ and $V$ to build tree $G$, one tree and two leaves disappear. Hence $\frac{1}{2}(2C_1)^2 = \frac{[\![U]\!][\![V]\!]}{[\![G]\!]} = \frac{\theta^2}{2}$. We get values of $C_k$ for $k > 1$ by applying Eq. (29) to a star graph $T_k = \{0 \leftrightarrow 1, 0 \leftrightarrow 2, \ldots, 0 \leftrightarrow k\}$ (Fig. 5, left).

$$[\![T_k]\!] = C_k \theta^k\,. \tag{31}$$

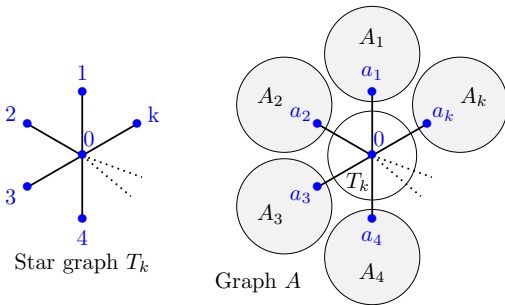

Figure 5: Left: the $k$ links of a star graph $T_k$ all are leaves with a common site $s = 0$. Right: graph $A$ used in Sec.4, where the $A_i$ subgraphs are any non-empty graph that do not include the link $a_i \leftrightarrow 0$. The $A_i$'s and the central star graph $T_k$ form a partition of the links of $A$.

It remains to prove Eq. (30). For this we consider a graph $A$ that possesses $k$ links originating from a site $0$, namely $0 \leftrightarrow a_1, 0 \leftrightarrow a_2, \ldots, 0 \leftrightarrow a_k$. These links may be either bridges or leaves. We denote $A_1, A_2, \ldots A_k$ the $k$ components of $A$ containing sites $a_1, a_2, \ldots a_k$, obtained by removing these links (see Fig. 5, right). Replacing in $[\![A]\!]$ every $P_{0 \leftrightarrow a_i}$ by $p_0^{++} p_{a_i}^{++} + p_0^{--} p_{a_i}^{--}$, we obtain (as we did to get Eq. (25)):

$$[\![A]\!] = \sum_{\varepsilon \in \{+,-\}^k} [\![p_0^{\varepsilon_1 \varepsilon_1}, \ldots, p_0^{\varepsilon_k \varepsilon_k}]\!] \prod_{i=1}^{k} [\![A_i]\!]_{a_i}^{\varepsilon_i} . \tag{32}$$

There we replace every $p_0^{--} = \mathbf{I} - p_0^{++}$ by $-p_0^{++}$ and get:

$$[\![A]\!] = [\![p_0^{++(k)}]\!] \prod_{i=1}^{k} ([\![A_i]\!]_{a_i}^{+} - [\![A_i]\!]_{a_i}^{-}) . \tag{33}$$

If all $A_i$'s are empty we get $[\![T_k]\!] = [\![(p_0^{++})^{(k)}]\!] \theta^k$. Hence $C_{k>1} = [\![(p_0^{++})^{(k)}]\!]$. This cumulant is the $k$th derivative of the logarithm of the following moment (see Eq. (B.4)):

$$\langle\!\langle e^{\lambda p_0^{++}} \rangle\!\rangle = \langle\!\langle e^{\lambda} p_0^{++} + p_0^{--} \rangle\!\rangle = \frac{e^{\lambda + h/2} + e^{-h/2}}{Y} .$$

Since $k > 1$, we get:

$$C_k = [\![p_0^{++(k)}]\!] = \frac{\partial^k}{\partial \lambda^k} \ln \langle\!\langle e^{\lambda p_0^{++}} \rangle\!\rangle \Big|_{\lambda=0} = \frac{\partial^k}{\partial \lambda^k} \ln \frac{e^{\lambda + \frac{h}{2}} + e^{-\frac{h}{2}}}{Y} \Big|_{\lambda=0} \tag{34}$$

$$= \frac{\partial^k}{\partial \lambda^k} \ln \cosh \frac{\lambda + h}{2} \Big|_{\lambda=0} = \frac{d^k}{dh^k} \ln \cosh \frac{h}{2} \tag{35}$$

$$= \frac{1}{2} \frac{d^{k-1}}{dh^{k-1}} \tanh \frac{h}{2} = \frac{d^{k-1}}{dh^{k-1}} \frac{\theta}{2} . \tag{36}$$

For $k = 1$, Eq. (34) gives the wrong value $\theta_+$, while Eq. (36) is equivalent to Eq. (30) for all $k \geq 1$. $\qquad \square$

Formulae (29) and (30) allow for a calculation in $O(n^2)$ of $F(T)$ for any tree $T$ with $n$ links, to be compared with the $O(4^n n^{3/2})$ of the method used for any graph in Sec. 2.5.

## 5  Discussion and conclusion

We have detailed the two steps involved in the exact calculation of the HTSE coefficients for Heisenberg $S = 1/2$ spin lattices, in the presence of a magnetic field: ($i$) the graph enumeration and ($ii$) the trace calculation. The trace calculations on bridged graphs (and particularly on trees) with $n$ links are the most time consuming steps, with a complexity in $O(n^{3/2}2^n)$ for a naive calculation. Formulae that drastically decrease it to $O(n^2)$ have been derived.

An optimized and parallelized code using this optimization is available [53], together with the coefficients of many series [30] (the orders obtained on some lattices with first neighbor interactions are given in Tab. 1 of App. D.4). The time required by this code for the two main steps (graph enumeration and trace calculations) is summarized in App. D.5 for various numbers of CPUs and for some simple models. The current code allows to also calculate HTSE for models with anisotropic interactions and Dzyaloshinskii-Moriya interactions.

This code was used on many models without magnetic field [25, 35, 36]. However, its interest stays in the possibility to explore high-**B**. This was done on the kagome antiferromagnet [24, 42]. High-**B** were used experimentally in [42] to shift away Schottky anomalies and isolate the low-temperature intrinsic kagome contribution to the specific heat, fitted by the entropy method at temperatures well below the convergence radius of the series.

This highlights the importance of extrapolation methods, just as Fig. 1 does: temperatures accessible by naive extrapolation techniques (Padé approximants) decrease with **B**. In case of magnetization plateaus, Padé approximants do not allow to reach temperatures where peaks (that are precursors of $T = 0$ plateaus) appear in $\chi_l(h)$. An extrapolation method designed to work in the $(T, \mathbf{B})$ plane would be a logical continuation of this work. To exploit the field dependent HTSE coefficients, thermodynamic ensembles other than the usual $(T, B)$ one could be considered, as evoked in Sec. 2.1.

Further studies could also extend this work to optimize HTSE calculation on a larger class of models (different spin values, classical models) in the presence of a magnetic field.

## Acknowledgments

**Funding information**  This work was supported by the French Agence Nationale de la Recherche under Grant No. ANR-18-CE30-0022-04 LINK and the projet Emergence, of the Paris city.

## A  Vocabulary on graphs

All the definitions below are illustrated on Fig. 2.

- Graphs where each link appears only once are called *simple graphs*, and graphs where multiple links are allowed are called *multi-graphs*.

- A graph is *connected* when a path exists between any two of its sites (it has only one connected component).

- The *degree* $d°s$ of a site $s$ is the number of links emanating from it.

- A *leaf* is a link with a site of degree one.

- A *bridge* is a link that is not a leaf and belongs to no simple loop. So it connects two otherwise not connected components. A graph with a bridge is said *bridged*.

- A *big leaf* is a generalization of a leaf. If not a leaf it is a bridge in company of one of the two components it separates, provided this component is free of leaves or bridges.

So no big leaf can include another one. That is why all big leaves are disjoint, except when there is only one bridge and no leaf. Then there are two big leaves sharing the only bridge and we must pretend there is only one big leaf. This way big leaves are always disjoint as needed. Let $N_{bf}$ be the total number of bridges and leaves. Let $N_L$ be the (pretended) number of big leaves. Then $\min(N_{bf}, 2) \leq N_L \leq N_{bf}$.

- An *islet* of a graph $G$ is a connected component of the graph obtained after cutting every bridge of $G$ and replacing it by two leaves.

## B Averages, moments and cumulants

The moment and cumulant of a multiset or list of operators $x_1, \ldots, x_k$ are:

$$\langle x_1, \ldots, x_k \rangle = \frac{\partial}{\partial \lambda_1} \cdots \frac{\partial}{\partial \lambda_k} \left\langle e^{\sum_{i=1}^k \lambda_i x_i} \right\rangle \Big|_{\lambda=0} \tag{B.1}$$

$$= \frac{1}{k!} \sum_{\sigma \in S_k} \left\langle \prod_{i=1}^k x_{\sigma(i)} \right\rangle, \tag{B.2}$$

$$[x_1, \ldots, x_k] = \frac{\partial}{\partial \lambda_1} \cdots \frac{\partial}{\partial \lambda_k} \ln \left\langle e^{\sum_{i=1}^k \lambda_i x_i} \right\rangle \Big|_{\lambda=0}. \tag{B.3}$$

For a single operator $x_1$, moment $\langle x_1 \rangle$, cumulant $[x_1]$ and average $\langle x_1 \rangle$ are equal. So we can use the notation $\langle . \rangle$ for both average and moment. Furthermore $\langle x^{(k)} \rangle = \langle x^k \rangle = [x^k] \neq [x^{(k)}]$ for $k > 1$ if $x^{(k)}$ denotes $k$ occurrences $x, \ldots, x$ of a same operator.

If $x_i = x_j$ in definitions B.1 and B.3 we can state $\mu = \lambda_i + \lambda_j$. Then $\partial \mu / \partial \lambda_i = \partial \mu / \partial \lambda_j = 1$. Hence we can replace $\lambda_i x_i + \lambda_j x_j$ by $\mu x_i$ and both $\partial \lambda_i$ and $\partial \lambda_j$ by $\partial \mu$. More simply we can remove the term $\lambda_j x_j$ in the sum and replace $\partial \lambda_j$ by $\partial \lambda_i$. In this way we have for instance:

$$[x_1^{(3)}, x_2, x_3^{(4)}] = \frac{\partial^3}{\partial \lambda_1^3} \frac{\partial}{\partial \lambda_2} \frac{\partial^4}{\partial \lambda_3^4} \ln \langle e^{\sum_{i=1}^3 \lambda_i x_i} \rangle \Big|_{\lambda=0}. \tag{B.4}$$

We now consider that $x$ is an operator corresponding to a link $x$ of a graph. Note that we use from now on the vocabulary of graphs using this operator-link correspondance, but what follows is valid for any set or multiset of operators. Hence if $G$ is a simple graph, i.e. a set of distinct links, we have the Maclaurin expansion:

$$\ln \langle \exp \sum_{x \in G} \lambda_x x \rangle = \sum_{U \in \mathbb{N}^G} \frac{[x^{(U(x))}, x \in G]}{\prod_{x \in G} U(x)!} \prod_{x \in G} \lambda_x^{U(x)}. \tag{B.5}$$

Here $U \in \mathbb{N}^G$ is a mapping from $G$ to $\mathbb{N}$. For each link $x \in G$, the integer $U(x)$ is its multiplicity in the multiset $\{\!\{x^{(U(x))}, x \in G\}\!\}$. So $U$ is any multigraph whose support is a part of $G$. We will simplify notations in this last equation and rewrite it:

$$\ln \langle \exp \lambda G \rangle = \sum_{U \in \mathbb{N}^G} \frac{[U]}{U!} \lambda^U = \sum_{k=1}^{\infty} \sum_{V \in G^k} \frac{[V]}{k!} \lambda^V. \tag{B.6}$$

Instead of summing over multisets of links, we may sum over tuples of links. But a multiset $U \in \mathbb{N}^G$ of $k = \#U = \sum_{x \in G} U(x)$ links appears $k!/U!$ times among tuples $V \in G^k$ of $k$ links. Similarly we have also

$$\langle \exp \lambda G \rangle = \sum_{U \in \mathbb{N}^G} \frac{\langle U \rangle}{U!} \lambda^U = 1 + \sum_{k=1}^{\infty} \sum_{V \in G^k} \frac{\langle V \rangle}{k!} \lambda^V. \tag{B.7}$$

The constant coefficients of these series in powers of $\lambda$ are $\langle \emptyset \rangle = \langle e^0 \rangle = 1$ and $[\emptyset] = \ln 1 = 0$. So the coefficients of either of these two formal series can be computed from the coefficients of the other one by

$$\sum_{U \in \mathbb{N}^G} \frac{\langle U \rangle}{U!} \lambda^U = 1 + \sum_{n=1}^{\infty} \frac{1}{n!} \left( \sum_{U \in \mathbb{N}^G} \frac{[U]}{U!} \lambda^U \right)^n, \tag{B.8}$$

$$\sum_{U \in \mathbb{N}^G} \frac{[U]}{U!} \lambda^U = - \sum_{n=1}^{\infty} \frac{1}{n} \left( 1 - \sum_{U \in \mathbb{N}^G} \frac{\langle U \rangle}{U!} \lambda^U \right)^n. \tag{B.9}$$

## B.1 Moments expressed as polynomials of cumulants

For a simple graph $U = \{x_1, \ldots, x_k\}$, the coefficient of $\lambda^U$ in Eq. (B.8) is

$$\langle U \rangle = \sum_{p \in \mathcal{Q}(U)} \prod_{G \in p} [G], \tag{B.10}$$

where $\mathcal{Q}(U)$ is the set of partitions of $U$. The divisions by $U! = 1$ and $G! = 1$ disappear, since graph $U$ and its part $G$ are simple. Furthermore the division by $n!$ disappears also because the product of the cumulants of the $n$ parts of a partition $p$ appears $n!$ times with reordered factors within $(\cdots)^n$.

To generalize this formula to multigraphs, we no more use partitions of sets of links, but partitions of set $\{1, \ldots, n\}$ so that links $x_i$ no longer need to be different:

$$\langle x_1, \ldots, x_n \rangle = \sum_{p \in \mathcal{Q}(n)} \prod_{q \in p} [x_r, r \in q]. \tag{B.11}$$

$\mathcal{Q}(n)$ is the set of partitions of set $\{1, \ldots, n\}$. Example:

$$\langle x_1, x_2, x_3 \rangle = [x_1, x_2, x_3] + [x_1, x_2][x_3] + [x_1, x_3][x_2] + [x_1][x_2, x_3] + [x_1][x_2][x_3].$$

## B.2 Cumulants expressed as polynomials of moments

For a simple graph $U = \{x_1, \ldots, x_k\}$ the coefficient of $\lambda^U$ in Eq (B.9) is

$$[U] = \sum_{p \in \mathcal{Q}(U)} g_0(\#p) \prod_{G \in p} \langle G \rangle. \tag{B.12}$$

When going from Eq. (B.8) to Eq. (B.10), the coefficient $1/n!$ disappears when multiplied by $n!$. Here $(-1)^{n-1}/n$ multiplied by $n!$ becomes $g_0(n) = (-1)^{n-1}(n-1)!$. For multigraphs, we have:

$$[x_1, \ldots, x_n] = \sum_{p \in \mathcal{Q}(n)} g_0(\#p) \prod_{q \in p} \langle x_r, r \in q \rangle. \tag{B.13}$$

Examples:

$$[x_1, x_2] = \langle x_1, x_2 \rangle - \langle x_1 \rangle \langle x_2 \rangle = \langle x_1 x_2 \rangle - \langle x_1 \rangle \langle x_2 \rangle,$$
$$[x_1, x_2, x_3] = \langle x_1, x_2, x_3 \rangle - \langle x_1, x_2 \rangle \langle x_3 \rangle - \langle x_1, x_3 \rangle \langle x_2 \rangle - \langle x_1 \rangle \langle x_2, x_3 \rangle + 2 \langle x_1 \rangle \langle x_2 \rangle \langle x_3 \rangle$$
$$= \frac{\langle x_1 x_2 x_3 \rangle + \langle x_1 x_3 x_2 \rangle}{2} - \langle x_1 x_2 \rangle \langle x_3 \rangle - \langle x_1 x_3 \rangle \langle x_2 \rangle - \langle x_1 \rangle \langle x_2 x_3 \rangle + 2 \langle x_1 \rangle \langle x_2 \rangle \langle x_3 \rangle.$$

## B.3 Moment and cumulants of a single operator

When $x_1 = x_2 = \cdots = x_n = H$, the equations (B.11) and (B.13) become

$$\langle H^n \rangle = \sum_{\substack{n_1,\ldots,n_n \in \mathbb{N}, \\ \sum_i i n_i = n}} n! \prod_i \frac{[H^{(i)}]^{n_i}}{(i!)^{n_i} n_i!}, \tag{B.14}$$

$$[H^{(n)}] = \sum_{\substack{n_1,\ldots,n_n \in \mathbb{N}, \\ \sum_i i n_i = n}} g_0 \left( \sum_i n_i \right) n! \prod_i \frac{\langle H^i \rangle^{n_i}}{(i!)^{n_i} n_i!}. \tag{B.15}$$

Examples:

$$[H] = \langle H \rangle,$$
$$[H^{(2)}] = \langle H^2 \rangle - \langle H \rangle^2,$$
$$[H^{(3)}] = \langle H^3 \rangle - 3 \langle H^2 \rangle \langle H \rangle + 2 \langle H \rangle^3,$$
$$[H^{(4)}] = \langle H^4 \rangle - 3 \langle H^2 \rangle^2 - 4 \langle H^3 \rangle \langle H \rangle + 12 \langle H^2 \rangle \langle H \rangle^2 - 6 \langle H \rangle^4,$$
$$[H^{(5)}] = \langle H^5 \rangle - 5 \langle H \rangle \langle H^4 \rangle - 10 \langle H^2 \rangle \langle H^3 \rangle + 20 \langle H^3 \rangle \langle H \rangle^2 + 30 \langle H \rangle \langle H^2 \rangle^2 - 60 \langle H^2 \rangle \langle H \rangle^3$$
$$+ 24 \langle H \rangle^5,$$
$$\langle H \rangle = [H],$$
$$\langle H^2 \rangle = [H^{(2)}] + [H]^2,$$
$$\langle H^3 \rangle = [H^{(3)}] + 3[H^{(2)}][H] + [H]^3,$$
$$\langle H^4 \rangle = [H^{(4)}] + 3[H^{(2)}]^2 + 4[H^{(3)}][H] + 6[H^{(2)}][H]^2 + [H]^4,$$
$$\langle H^5 \rangle = [H^{(5)}] + 15[H][H^{(2)}]^2 + 10[H^{(2)}][H]^3 + [H]^5 + 5[H][H^{(4)}] + 10[H^{(2)}][H^{(3)}]$$
$$+ 10[H^{(3)}][H]^2.$$

## B.4 Expression of cumulants versus moments and lesser order cumulants

From Eq. (B.10) we can easily derive, if $x_1 \in X$:

$$\langle X \rangle = \sum_{X' \subset X \setminus x_1} [X \setminus X'] \langle X' \rangle. \tag{B.16}$$

Hence

$$[x_1, \ldots, x_n] = \langle x_1, \ldots, x_n \rangle - \sum_{p \in P_2''(n)} [x_r, r \in p_1] \langle x_r, r \in p_2 \rangle, \tag{B.17}$$

where $P_2''(n)$ is the set of partitions of $n$ elements in 2 non-empty sets, with the conditions that 1 is in the first set.
Example:

$$[x_1, x_2, x_3] = \langle x_1, x_2, x_3 \rangle - [x_1, x_2] \langle x_3 \rangle - [x_1, x_3] \langle x_2 \rangle - [x_1] \langle x_2, x_3 \rangle.$$

## B.5 Nullity of cumulant of a not connected graph

Let $C$ be a not connected graph, without any isolated site. Let $A$ be one of its connected component. Let $B = C \setminus A$. Then $A$ and $B$ are two non-empty graphs sharing no sites and operators $\lambda A = \sum_{x \in A} \lambda_x x$ and $\lambda B$ are independent. So their exponentials are independent too:

the average of their product is the product of their averages. But as $\lambda C = \lambda A + \lambda B$, we have $\ln\langle \exp \lambda C\rangle = \ln\langle \exp \lambda A\rangle + \ln\langle \exp \lambda B\rangle$. So

$$\sum_{U\in\mathbb{N}^C} \frac{[U]}{U!}\lambda^U = \sum_{U\in\mathbb{N}^B} \frac{[U]}{U!}\lambda^U + \sum_{U\in\mathbb{N}^A} \frac{[U]}{U!}\lambda^U. \tag{B.18}$$

If $U$ is a multigraph of support $C$, the term $[U]\lambda^U/U!$ appears only once in Eq. (B.18) in its left hand side. No other term has the same $\lambda^U$. Hence $[U] = 0$, which proves that the cumulant of a not connected multigraph is zero.

## B.6 Multilinearity of moments and cumulants

With Eq. (B.2) we see that the moment is a linear function of any of its arguments. Then with Eq. (B.13) we see that the cumulant is also such a function.

## B.7 Product of cumulants of independent sets of operators

Let $X$ and $Y$ be two sets of operators. Let $x_1 \in X$ and $y_1 \in Y$.

$$(\forall X' \subset X,\ \forall Y' \subset Y,\ \langle X', Y'\rangle = \langle X'\rangle\langle Y'\rangle) \quad \Rightarrow \quad [X][Y] = [x_1 y_1, X \setminus x_1, Y \setminus y_1]. \tag{B.19}$$

*Proof.* We prove this by induction on $\#X + \#Y$. We denote $X_1 = X \setminus x_1$ and $Y_1 = Y \setminus y_1$. We have $\langle X\rangle\langle Y\rangle = \langle X, Y\rangle = \langle x_1 y_1, X_1, Y_1\rangle$. Hence using three times Eq. (B.16):

$$\left(\sum_{X'\subset X_1}[X \setminus X']\langle X'\rangle\right)\left(\sum_{Y'\subset Y_1}[Y \setminus Y']\langle Y'\rangle\right) = \sum_{W'\subset X_1\cup Y_1}[x_1 y_1, X_1 \cup Y_1 \setminus W']\langle W'\rangle,$$
$$\sum_{\substack{X'\subset X_1 \\ Y'\subset Y_1}}[X \setminus X'][Y \setminus Y']\langle X'\rangle\langle Y'\rangle = \sum_{\substack{X'\subset X_1 \\ Y'\subset Y_1}}[x_1 y_1, X_1 \setminus X', Y_1 \setminus Y']\langle X'\rangle\langle Y'\rangle.$$

According to the induction hypotheses, all terms for $X' \neq \emptyset$ or $Y' \neq \emptyset$ cancel. The remaining terms are those of Eq. (B.19). □

# C Proof of the non contribution of some graphs in the fixed-$B$ expansion

If $U$ is a connected multigraph with $N_L$ big leaves then

$$\theta^{N_L} \text{ divides } [\![U]\!]. \tag{C.1}$$

*Proof.* We assume that a multiple link cannot be a leaf or a bridge. Let $k = N_L$. Let $A_1, \ldots, A_k$ be the parts of $U$ which are disconnected when removing the leaves or bridges of the big leaves. Let $B = U \setminus A_1 \setminus A_2 \setminus \cdots \setminus A_k$. A big leaf is $A_i \cup \{a_i \leftrightarrow b_i\}$ with $a_i$ in $A_i$ and $b_i$ in $B$. Then, the very same proof as for Eq. (33) gives:

$$[\![U]\!] = [\![B, p_{b_1}^{++}, \ldots, p_{b_k}^{++}]\!]\prod_{i=1}^{k}([\![A_i]\!]_{a_i}^{+} - [\![A_i]\!]_{a_i}^{-}). \tag{C.2}$$

Replacing $\theta$ by $-\theta$ in $[\![A_i]\!]_{a_i}^{+}$ gives $[\![A_i]\!]_{a_i}^{-}$. Hence $[\![A_i]\!]_{a_i}^{+} - [\![A_i]\!]_{a_i}^{-}$ is an odd polynomial in $\theta$ and it is divisible by $\theta$. □

## C.1  Graphs with $\#G + N_{bf} > n$ do not contribute for $B = 0$

Here, we prove the first item of Sec. 2.4.2: for $B = 0$, a simple connected graph $G$ with $N_{bf}$ links that are either bridges or leaves can be discarded if $\#G + N_{bf} > n$. We recall that $N_{bf} \geq N_L$, with $N_L$ the number of big leaves, see App. A.

*Proof.* The reason is that in this case, $F(G) = o(\beta^n)$. Let $U$ be a multi-graph $U$ of support $G$. If $\#U \geq \#G + N_{bf}$, then $\#U > n$ and $J^U = o(\beta^n)$. Otherwise $\#U < \#G + N_{bf}$. Doubling $\#U - \#G$ links disables at most as many bridges or leaves. But at least one remains. Hence $U$ has a big leaf, and $[\![U]\!]$ is divisible by $\theta$, meaning since $\theta = 0$ that $[\![U]\!] = 0$. $\qquad\square$

## C.2  Graphs with $\#G + N_L > n$ do not contribute to the fixed-$B$ expansion

Now we count only leaves and bridges inside big leaves to prove the second item of Sec. 2.4.2: A graph $G$ with $N_L$ big leaves does not contribute to the fixed-$B$ expansion at order $n$ if $\#G + N_L > n$. Let $U$ be a multi-graph of support $G$. Then $\theta^{N_L(U)}$ divides $[\![U]\!]$. Hence

$$\text{order}_\beta \frac{J^U [\![U]\!]}{U!} \geq \#U + N_L(U) \geq \#G + N_L(G) > n. \tag{C.3}$$

## C.3  More restrictive criterion for the fixed-$B$ expansion

We now explain a criterion (C.4) that allows to discard more graphs than (C.3), and we give an algorithm to compute it.

To write this criterion, we define odd islets. In a connected multigraph $U$ with $N_b$ bridges, we can replace every bridge $l = l_1 \leftrightarrow l_2$ by two leaves $l_1 \leftrightarrow l_4$ and $l_3 \leftrightarrow l_2$ where $l_3$ and $l_4$ are new sites. We get $N_b + 1$ connected components, that we call *islets* (see App. A) and denote $U_0, U_1 \dots U_{N_b}$. An islet with an odd number of leaves (including broken bridges) is said to be *odd*. We denote $N_f$ and $N_o$ the numbers of leaves and odd islets and we define $N_{fo} = N_f + N_o$. Note that $N_l$ is the number of links, but we often replace it by $\#$ as $\#G = N_l(G)$.

The new criterion to discard a simple connected graph $G$ for a fixed-$B$ expansion writes:

$$n < \min_{U \in \{1,2\}^G} \left( \#U + N_{fo}(U) \right), \tag{C.4}$$

where $U$ are multigraphs of support $G$ where links have multiplicities 1 or 2.

*Proof.* Eq. (C.1) tells us that $\theta^{N_f(G)}$ divides $[\![G]\!]$ and $\theta^{N_f(U_i)}$ divides $[\![U_i]\!]$. This is coherent with $N_f(G) = \sum_{i=0}^{N_b} N_f(U_i) - 2N_b$ and $[\![G]\!] = (2/\theta^2)^{N_b} \prod_i [\![U_i]\!]$. But $[\![U_i]\!]$ is an even polynomial of $\theta$. So when $U_i$ is an odd islet, $\theta^{N_f(U_i)+1}$ divides $[\![U_i]\!]$. This proves that

$$\theta^{N_{fo}(G)} \stackrel{\text{def}}{=} \theta^{N_f(G)+N_o(G)} \text{ divides } [\![G]\!]. \tag{C.5}$$

This improves criterion (C.1), since big leaves are leaves and islets with one leaf and $N_{fo} \geq N_L$.

In Eq. (C.5) we can replace simple graph $G$ by a multigragh of support $G$. However when doubling a bridge between two odd islets, they are disabled and replaced by a single even islet. And doubling a leaf of an odd islet disables the leaf and the odd islet. So $N_{fo}$ may decrease by two when doubling a link. This is why we have only $F(G) = O(\beta^{N_l + (N_{fo})/2})$ and we can discard a graph $G$ when $n < N_l + \frac{N_{fo}}{2}$, or better when combined with Sec. C.2:

$$n < N_l + \max\left( N_L, \frac{N_{fo}}{2} \right). \tag{C.6}$$

This is the best possible criterion, if we use only $N_l$, $N_L$, $N_f$ and $N_o$. But the real criterion to discard a simple connected graph $G$ is to make sure that $n < \#U + N_{fo}(U)$ for every multigraph $U$ of support $G$. Since $\#U$ increases and $N_{fo}(U)$ does not change when we increase the multiplicity of an already multiple link, we can limit multiplicities to 2. This is Eq. (C.4). $\qquad\square$

At first glance the time to evaluate this formula is $O(2^{N_l} N_l)$. It reduces to $O(2^{N_{bf}} N_l)$ if we notice that only leaves and bridges are worth doubling. We may also notice that $\#U + N_{fo}(U)$ increases by 0 or 2 when we double two bridges leading to a same islet. Hence we forbid this. This leads to:

**Algorithm for Eq. (C.4):** Minimal $U$ can be found in time $O(N_l)$. Since Eq. (C.4) uses total number of leaves and odd islets, we will make no difference between a leaf and a big leaf. Hence a leaf will be called a bridge and the site at the end of this leaf will be called an (odd) islet. We start from $U = G$. Then as long as $U$ has a bridge between two odd islets and one of these islets has no other bridge to a third odd islet, we double this bridge. Note that after doubling, this bridge is no longer a bridge and the two odd islets become a single even islet. Hence $\#U + N_{fo}(U)$ decreases by 1. This walk through $\{1,2\}^G$ ends when there is no more bridge to double. Then $U$ is a local minimum. But we chose an outermost bridge (no third odd islet) to insure that disabled bridges are all in the same side of the doubled bridge. Hence the chosen bridge excludes at most one other bridge of any optimal solution. Then it will replace it in this solution, yielding another optimal solution which is reachable. All of this means that at the end of its walk, $U$ is indeed a global minimum. In other words, condition "no third odd islet" avoids being stuck in a local minimum. For instance, starting from $G = a \leftrightarrow b \leftrightarrow c \leftrightarrow d$, where all the bridges are drawn and letters stand for islets, we cannot be stuck in $a \leftrightarrow b \Leftrightarrow c \leftrightarrow d$ whereas mimimum is $a \Leftrightarrow b \leftrightarrow c \Leftrightarrow d$.

Searching for a bridge to double and doubling it, takes time $O(N_l)$. No more than $N_l/2$ links are doubled. Hence the total time is $O(N_l^2)$. But algorithm can be performed in time $O(N_l)$. For this we first shrink each islet in a single site. This turns $G$ into a tree. Then we compute the oddness of every islet. After that, a depth first search on the tree finds which links to double: When backtracking through a link, if both its ends are (still) odd, this link is doubled and its ends become even.

## C.4 Criteria for $F(G) = o(J^n) + o(\theta^\nu)$

We may want to compute $g_\infty + o(J^n) + o(\theta^\nu)$ instead of $g_\infty + o(\beta^n)$. Then criteria (C.6) and (C.4) to discard $G$ become

$$n < N_l \quad \vee \quad n + \nu < N_l + \max\left(N_L, \frac{N_{fo}}{2}\right), \tag{C.7}$$

$$n < N_l \quad \vee \quad \nu < \min_{\substack{U \in \{1,2\}^G \\ \#U \le n}} N_{fo}(U). \tag{C.8}$$

Then minimal $U$ is harder to find. We first transform the graph $G$ into a rooted tree, by keeping only bridges and leaves and replacing every islet by a single site and choosing a root. From now on, an islet means either an islet or the end of a leaf.

We define the potential of a rooted tree $T$ with $k$ links, as

$$\text{pot}(T) = (u, v) = ((u_0, u_1, \ldots, u_k), (v_0, v_1, \ldots, v_k)),$$

where $u_i$ (resp. $v_i$) denotes the minimum of $N_{fo}(U)$ for $U \in \{1,2\}^T$ with $\#U = k + i$ and the root of $T$ being in an even (resp. odd) islet (or site) of $U$. For instance $u_k = 0$, $v_k = \infty$ (as all islets are even for $\#U = 2k$) and $\{u_0, v_0\} = \{N_{fo}(T), \infty\}$, where $\infty$ stands for the minimum of an empty set.

So if the only common site of trees $T$ and $T'$ is their root and $\text{pot}(T) = (u, v)$ and $\text{pot}(T') = (u', v')$ then $\text{pot}(T \cup T') = (\min(u \oplus u', v \oplus v' - 2), \min(u \oplus v', v \oplus u'))$, where $(a \oplus b)_i = \min_{i=i'+i''} a_{i'} + b_{i''}$.

Furthermore if $T' = T \cup a \leftrightarrow a'$ and $a$, resp $a'$, is the root of $T$, resp. $T'$, and $\mathrm{pot}(T) = (u, v)$ then $\mathrm{pot}(T') = (\infty\widehat{\phantom{x}}u, \min(\infty\widehat{\phantom{x}}v, 1 + u\widehat{\phantom{x}}\infty, 1 + v\widehat{\phantom{x}}\infty))$ where $\infty\widehat{\phantom{x}}(u_0, u_1, u_2) = (\infty, u_0, u_1, u_2)$. Using these two operations and starting from $\mathrm{pot}(\emptyset) = ((0), (\infty))$ or $\mathrm{pot}(a \leftrightarrow b) = ((\infty, 0), (1, \infty))$, we can build any rooted tree and its potential in time $O(N_l^3)$. If $\mathrm{pot}(T) = (u, v)$ then Eq. (C.8) reads $n < N_l \lor v < \min(u_{n-N_l}, v_{n-N_l})$.

# D  Proof of some complexities

In the three following subsections, the complexities of the three successive steps listed in Sec. 2.5 are detailed.

## D.1  Moments

A simple (not so naive) way to calculate the moments $\langle\langle H_J^k \rangle\rangle$ for all $k \leq n$ on a graph $G$ is to work in the basis of up and down spin in the $z$ direction, of size $2^{N_s}$. It sub-divides into sectors of fixed magnetization $m = S^z$, from $-N_s/2$ to $N_s/2$ by integer steps (see Algorithm 1). The basis vectors are denoted $|v_{i,m}\rangle$ or simply $|v_i\rangle$ when $m$ depends on $i$. The traces are calculated separately in each subsector: $\mathrm{Tr}_m H_J^k = \sum_i \langle v_{i,m} | H_J^k | v_{i,m} \rangle$. We get $\langle\langle H_J^n \rangle\rangle$ by summing them with the appropriate weight:

$$\langle\langle H_J^k \rangle\rangle = \sum_{m=-N_s/2}^{N_s/2} \frac{e^{hm}}{Y^{N_s}} \mathrm{Tr}_m H_J^k = \sum_{m'=0}^{N_s} \theta_+^{m'} \theta_-^{N_s - m'} \mathrm{Tr}_{m' - \frac{N_s}{2}} H_J^k. \tag{D.1}$$

The partial traces $\mathrm{Tr}_m H_J^k$ for any $k \leq n$ are obtained by first calculating $|v_{i,m}^{(1)}\rangle = H_J |v_{i,m}\rangle$, then $|v_{i,m}^{(2)}\rangle = H_J |v_{i,m}^{(1)}\rangle$ and so on up to $|v_{i,m}^{(n)}\rangle$. Then, we get $\mathrm{Tr}_m H_J^k = \sum_i \langle v_{i,m}^{(k)} | v_{i,m} \rangle$ for $k \leq n$. We may also compute $\mathrm{Tr}_m H_J^k = \sum_i \langle v_{i,m}^{(\lceil k/2 \rceil)} | v_{i,m}^{(\lfloor k/2 \rfloor)} \rangle$ for $k \leq n$, where $\lceil . \rceil$ and $\lfloor . \rfloor$ are the ceiling and floor functions. So we need $|v_{i,m}^{(k)}\rangle$ only up to $k = \lceil n/2 \rceil$ and computation is twice as fast and involves smaller intermediate numbers. The complexity of the naive calculation of all $\langle\langle H_J^k \rangle\rangle$, $k \leq n$ is $O(4^{N_s} n N_l)$, as we have to calculate the $2^{N_s}$ coefficients of the image of $2^{N_s}$ basis vectors, $n/2$ times (for each power of $H_J$), with an extra factor $N_l$, because $H_J$ is a sum of $N_l$ simple operators. The result is an even polynomial in $\theta = \tanh \frac{h}{2}$ of maximal order $N_s$: we group terms with opposite magnetization $m$ and $-m$, to get a weight proportional to $\frac{\cosh mh}{Y^{N_s}}$, which is an even polynomial in $\theta$ of degree $N_s$ (when all $J$'s are identical and $H_J$ is divided by $J$, the coefficients of this polynomial are simple numbers, and not polynomials in $J_l$'s, which would increase the complexity). The degree in $\theta$ of $\langle\langle H_J^k \rangle\rangle$ is in fact $\min(N_s, 2k)$, as a term of $H_J^k$ corresponds to a set of $k$ links. Whatever the set, a maximum number of $2k$ sites appear. The other sites are free and do not influence the average for this term.

In algorithm 1 we may skip iterations of loop **for** $i \dots$ when $m < 0$ and supply missing values in array $t$ by $t[k, m'] = t[k, N_s - m']$ for $m' < N_s/2$. This saves half the computation time.

If we store $\langle v_j | v \rangle$ for all $j \in \{+, -\}^{N_s}$ in an array of $2^{N_s}$ integers, it is easy to perform $|w\rangle += P_l |v\rangle$ in time $O(2^{N_s})$. But most of these integers are zeros. Handling only the relevant components, those for which $j$ has same magnetization as $i$, is tricky but reduces time to $O(\binom{N_s}{m'})$. So in the overall estimated time of algorithm 1, factor $4^{N_s}$ is replaced by $\sum_{m'=0}^{N_s} \binom{N_s}{m'}^2 = \binom{2N_s}{N_s} \sim \frac{4^{N_s}}{\sqrt{N_s \pi}}$. The time is divided by $\sqrt{N_s \pi}$ and becomes $O(4^{N_s} n N_l / \sqrt{N_s})$. In C language a simple trick could be to replace the loop

```
        for(j=0 ; j<1<<Ns ; j++)
```
by
```
        for(j=(1<<__builtin_popcount(i))-1 ; j<1<<Ns ;
```

---

**Algorithm 1:** Calculation of $\overline{g}(G)$ and $g(G)$

---

$\quad$ **for** $k$ from 1 to $n$, $m'$ from 0 to $N_s(G)$ **do**
$\quad\quad t[k,m'] = 0$
$\quad$ **end for**
$\quad$ **for** $i$ in $\{+,-\}^{N_s(G)}$ **do**
$\quad\quad m' = $ number of $+$ in $i$
$\quad\quad |v\rangle = |v_i\rangle$ $\qquad\qquad\qquad\qquad$ // of magnetization $m = m' - \frac{N_s}{2}$
$\quad\quad$ **for** $k$ from 1 to $n$ **do**
$\quad\quad\quad |w\rangle = 0$
1 $\quad\quad\quad$ **for** $l$ in $G$ **do**
$\quad\quad\quad\quad |w\rangle \mathrel{+}= P_l |v\rangle$ $\qquad\qquad\qquad$ // $O(2^{N_s(G)})$ or $O(\binom{N_s}{m'})$
$\quad\quad\quad$ **end for**
$\quad\quad\quad |v\rangle = |w\rangle$
$\quad\quad\quad t[k,m'] \mathrel{+}= \langle v_i | v\rangle$ $\qquad\qquad\qquad$ // $O(1)$
$\quad\quad$ **end for**
$\quad$ **end for**
$\quad \overline{g} = 1 + \sum_{m'} \theta_+^{m'} \theta_-^{N_s - m'} \sum_{k=1}^{n} \frac{J^k}{k!} t[k,m']$ $\qquad$ // $O(nN_s^2)$
$\quad g = -\sum_{i=1}^{n} \frac{(1-\overline{g})^i}{i}$ $\qquad\qquad\qquad\qquad$ // $O(n^4 N_s)$

---

$$\texttt{j+=a=j\&-j,j+=((j\&-j)>>\_\_builtin\_ctz(a+a))-1)}$$

where $\texttt{j}$ jumps efficiently to the next integer value with a same number of ones in binary as $\texttt{i}$.

But there is a simpler way which divides the time by only $\sqrt{N_s \pi / 2}$. Instead of computing $|v_{i,m}^{(k)}\rangle = H_j^k |v_{i,m}\rangle$, we compute $|V_i^{(k)}\rangle = H_j^k |V_i\rangle$ with $|V_i\rangle = \sum_m |v_{i,m}\rangle$. Of course $|v_{i,m}\rangle = 0$ if $i$ is too big. Then components of various magnetizations do not mix, and we get $\langle v_{i,m}^{(k)} | v_{i,m}\rangle = \langle V_i^{(k)} | v_{i,m}\rangle$. This way, instead of computing $|v_i^{(k)}\rangle$ for $2^{N_s}$ values of $i$, we compute $|V_i^{(k)}\rangle$ for $\binom{N_s}{\lfloor N_s/2 \rfloor}$ values of $i$.

We can still save half the computational time thanks to spin reversal. Assuming that reversing spins in $|v_{i,m}\rangle$ gives $|v_{A_m - i, -m}\rangle$, we have $\langle v_{i,m}^{(k)} | v_{i,m}\rangle = \langle V_{A_m - i}^{(k)} | v_{A_m - i, -m}\rangle$, where $A_m = \binom{N_s}{m + N_s/2} + 1$. So we need $|V_i^{(k)}\rangle$ for only half as many values of $i$.

Furthermore we can save about half computation time in Algorithm 1 if we replace $|w\rangle = 0$ by $|w\rangle = N_l |v\rangle$ and $|w\rangle \mathrel{+}= P_l |v\rangle$ by $|w\rangle \mathrel{+}= (P_l - \mathbf{I}) |v\rangle$, since

$$P_l - \mathbf{I} = (|S_{l+-}^z\rangle - |S_{l-+}^z\rangle)(\langle S_{l-+}^z| - \langle S_{l+-}^z|), \qquad\qquad\text{(D.2)}$$
$$P_l = |S_{l++}^z\rangle \langle S_{l++}^z| + |S_{l--}^z\rangle \langle S_{l--}^z| + |S_{l+-}^z\rangle \langle S_{l-+}^z| + |S_{l-+}^z\rangle \langle S_{l+-}^z|,$$

where $\langle S_{l\epsilon\epsilon'}^z| = \langle S_{l_1}^z = \epsilon, S_{l_2}^z = \epsilon'|$.

## D.2 Logarithm expansion

Going from the series of moments $\overline{g}(G)$ of Eq. (10) to the series of cumulants $g(G)$ of Eq. (11) requires the expansion of the logarithm up to order $n$ in $J$. In the calculation $g = -\sum_{i=1}^{n}(1-\overline{g})^i/i$, all the powers of $1-\overline{g}$ and the result $g$ are polynomials of degree $n$ in $J$ where the coefficient of $J^k$ is an even polynomial of maximal degree $2k$ in $\theta$. They have $\sim n^2/2$ integer coefficients (of $J^k \theta^{2i}/(k!2^k)$ for $i \leq k \leq n$) see 2.3. The complexity of this step with $n$ multiplications of such polynomials is $O(n(n^2)^2) = O(n^5)$, or better $O(n^4 N_s)$ since the first multiplicand is always $1-\overline{g}$ with only $O(nN_s)$ non zero coefficients, since $d_\theta^\circ \overline{g} \leq N_s$. Moreover $d_\theta^\circ (1-\overline{g})^i \leq 2i\lfloor N_s/2 \rfloor$ and the coefficient of $J^k$ in $(1-\overline{g})^i$ is a polynomial in $\theta^2$ of degree $\min(k, i\lfloor N_s/2 \rfloor)$.

Before this calculation we must transform $\overline{g}$ which is implicitly contained in the matrix of integers $t$ (defined in Algorithm 1) into an explicit polynomial in $J$ and $\theta$. The computation of its coefficients costs a time in $O(nN_s^2)$.

### D.3  Calculation of $F(G)$

For the last step, we suppose that we know all the $F(G')$ for $G'$ smaller than $G$. In a naive evaluation of eq.(15), the connectivity of each $G'$ among the $2^{N_l}$ subsets of $G$ is checked in time $O(N_l)$ and if needed we add polynomial $F(G')$ of degree $n$ in $J$ and $\theta^2$ in time $O(n^2)$. The complexity of this step is $O(n^2 2^{N_l})$, that we reduce to $O(n^2 N_l^2)$ as explained now. To avoid the graph enumeration, we are tempted to replace the sum of Eq. (15) by a sum over graphs $G'$ obtained from $G$ by removing a single link. We face the problem that graphs included in $G \setminus \{l, l'\}$ are at least in both $G \setminus \{l\}$ and $G \setminus \{l'\}$, and must not be counted several times. We group the $F(G')$ having graphs with the same number of links $i$ into $\breve{F}_i(G)$:

$$\breve{F}_i(G) = \sum_{\substack{G' \subset G, \\ N_l(G')=i}} F(G'). \tag{D.3}$$

Now $\breve{F}_i(G)$ and $\breve{F}_i(G \setminus \{l\})$ are related through:

$$(N_l - i)\breve{F}_i(G) = \sum_{l \in G} \breve{F}_i(G \setminus \{l\}), \tag{D.4}$$

which gives $\breve{F}_i(G)$ for $i < N_l$. Then $F(G)$ is given by:

$$F(G) = \breve{F}_{N_l}(G) = g(G) - \sum_{i=1}^{N_l-1} \breve{F}_i(G). \tag{D.5}$$

If we know $\breve{F}_i(G')$ for all connected sub-graph $G' \subsetneq G$, we get $F(G)$ (and all the $\breve{F}_i(G)$'s) in a time $O(n^2 N_l^2)$: Eq. (D.4) needs calculating $N_l$ sums of $N_l$ polynomials with $\sim n^2/2$ coefficients (of $J^k \theta^{2j}$ for $j \leq k \leq n$). However, we have to consider that $\breve{F}_i(G \setminus \{l\})$ is not directly known when $G \setminus \{l\}$ is not connected. Then, it contains 2 connected components $G_1$ and $G_2$, and we get from Eq. (D.3) that $\breve{F}_i(G) = \breve{F}_i(G_1) + \breve{F}_i(G_2)$, which does not change the previously calculated complexity (see Alg. 2).

---

**Algorithm 2:** Calculation of $F(G)$

> **for** $i$ from 0 to $\#G$ **do**
>   $\breve{F}_i(G) = 0$
> **end for**
> **for** $l \in G$ **do**
>   **for** $G'$ connected component of $G \setminus \{l\}$ **do**
>     **for** $i$ from 1 to $\#G'$ **do**
>       $\breve{F}_i(G)$ += $\breve{F}_i(G')$         // $O(n^2)$
>     **end for**
>   **end for**
> **end for**
> **for** $i$ from 1 to $\#G - 1$ **do**
>   $\breve{F}_i(G)$ /= $\#G - i$
> **end for**
> $F(G) = \breve{F}_{\#G}(G) = g(G) - \sum_{i=1}^{\#G-1} \breve{F}_i(G)$

---

Table 1: Current available HTSE for $S : \frac{1}{2}$ Heisenberg model with first neighbor interactions. $n_l$ is the number of links per unit cell. $n_s$ is the number of sites per unit cell. $n$ and $n_Z$ are respectively the orders in $\beta$ and in $Z = \theta^2 = \tanh\frac{\beta h}{2}$. ssc means semi-simple cubic (see [59] for a description). The series are given in [30].

| model | $n_l$ | $n_s$ | $n$ ($n_Z = 0$) | $n$ ($n_Z = 1$) | $n$ ($n_Z = n$) |
|---|---|---|---|---|---|
| **1D** | | | | | |
| chain | 2 | 1 | 28 | | 18 |
| sawtooth | 3 | 2 | | 15 | |
| **2D** | | | | | |
| checkerboard | 6 | 2 | 18 | | 16 |
| honeycomb | 3 | 2 | 22 | 20 | 18 |
| kagome | 6 | 3 | 20 | 18 | 16 |
| square | 2 | 1 | 20 | 18 | 16 |
| triangular | 3 | 1 | 18 | | 16 |
| **3D** | | | | | |
| bcc | 4 | 1 | | 15 | 12 |
| fcc | 6 | 1 | | 13 | |
| Hyperkagome | 24 | 12 | 18 | | |
| Pyrochlore | 12 | 4 | | | 17 |
| sc | 3 | 1 | | 17 | 14 |
| ssc | 6 | 4 | | 20 | |

## D.4 Available HTSE

Series obtained with our algorithm are publicly available [30]. Their orders in $\beta$ and in $Z$ are recapitulated for several lattices in Tab. 1 for Heisenberg first neighbor interactions.

## D.5 Computation times

Benchmarks have been realized on AMD CPU's, whose times are recapitulated in Tab. 2. The order of the series in $\beta$: $n$, in $Z$: $n_Z$ are varied for several lattices, the number of cores used is indicated, and the computation time of the graph enumeration and of the trace calculation are given in seconds. The number $\#\{\overline{G}\}$ of graph classes with $n$ links and requiring a trace calculation is indicated. Note the variation depending on the graph coordination number $z$: this number of graph classes is similar at order 16 on the kagome and square lattice with $z = 4$, but much larger on the triangular one ($z = 6$).

Table 2: Comparison of computation time for some HTSE calculations, depending on the number of cores. Durations $t$ are in seconds. $n$ and $n_Z$ are respectively the orders in $\beta$ and in $Z = \theta^2 = \tanh \frac{\beta h}{2}$. The last columns indicates the number of contributing graph classes at last order, whose trace has to be calculated. Computations have been done on AMD-workstations (AMD EPYC 7702P 64-Core Processor).

| Lattice | $n$ | $n_Z$ | $n_{\text{cores}}$ | $t$ (graphs) | $t$ (traces) | $\#\{\overline{G}\}$ |
|---------|-----|-------|--------------------|--------------|--------------|----------------------|
| Square  | 16  | 0     | 1                  | 58           | 464          | 184                  |
|         | 16  | 0     | 2                  | 45           | 233          | 184                  |
|         | 16  | 0     | 4                  | 32           | 117          | 184                  |
|         | 16  | 0     | 8                  | 22           | 59           | 184                  |
|         | 16  | 0     | 16                 | 15           | 35           | 184                  |
|         | 16  | 0     | 32                 | 13           | 26           | 184                  |
|         | 16  | 0     | 64                 | 12           | 27           | 184                  |
|         | 16  | 1     | 16                 | 14           | 1521         | 7067                 |
|         | 16  | 1     | 32                 | 13           | 758          | 7067                 |
|         | 16  | 1     | 64                 | 12           | 650          | 7067                 |
|         | 16  | 16    | 16                 | 14           | 28750 (8h)   | 168119               |
|         | 16  | 16    | 32                 | 13           | 15246 (4h)   | 168119               |
|         | 16  | 16    | 64                 | 12           | 18994 (5h)   | 168119               |
| Triangle| 14  | 0     | 16                 | 305          | 8            | 3390                 |
|         | 14  | 0     | 32                 | 261          | 4            | 3390                 |
|         | 14  | 0     | 64                 | 271          | 3.4          | 3390                 |
|         | 14  | 1     | 16                 | 291          | 146          | 50849                |
|         | 14  | 1     | 32                 | 261          | 79           | 50849                |
|         | 14  | 1     | 64                 | 270          | 62           | 50849                |
|         | 14  | 14    | 16                 | 294          | 977          | 242352               |
|         | 14  | 14    | 32                 | 262          | 527          | 242352               |
|         | 14  | 14    | 64                 | 271          | 403          | 242352               |
| Kagome  | 16  | 0     | 16                 | 29           | 43           | 240                  |
|         | 16  | 0     | 32                 | 26           | 25           | 240                  |
|         | 16  | 0     | 64                 | 24           | 28           | 240                  |
|         | 16  | 1     | 16                 | 29           | 2012         | 10278                |
|         | 16  | 1     | 32                 | 26           | 1002         | 10278                |
|         | 16  | 1     | 64                 | 23           | 863          | 10278                |
|         | 16  | 16    | 16                 | 29           | 27645 (7.7h) | 198609               |
|         | 16  | 16    | 32                 | 26           | 14435 (4h)   | 198609               |
|         | 16  | 16    | 64                 | 23           | 17215 (5h)   | 198609               |

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
