# Peer review of "High temperature series expansions of S = 1/2 Heisenberg spin models: algorithm to include the magnetic field with optimized complexity"

_SciPost Physics, doi:SciPost Phys. 17, 105 (2024)_

## Round 1 · Referee Report · Anonymous (Referee 1) · 2024-4-19

Strengths

1) Detailed, step by step presentation of a new algorithm for calculation of the high-temperature series for the general spin-1/2 exchange Hamiltonian with the Zeeman term

Weaknesses

1) introduction is too short and technical 2) no actual results are presented

Report

This is an important theoretical work which extends capability of the high-temperature expansion technique for quantum spin models by including the effect of strong magnetic field. Still a couple of improvements on the manuscript can be made that will increase visibility of the current work within quantum magnetism community. The introduction is extremely short and a bit out of point. Instead of a somewhat ambiguous statement about general validity of the Hubbard model (not studied in this work) authors may give a brief historical overview of the high-temperature expansion methods. For deeper appreciation of their results, authors may also include explicit results for one or two simple spin models. For example, the series obtained for square and kagome lattice antiferromagnets can be used to compute the uniform magnetization M(T,h) for a few values of h ~ J.

Recommendation

Ask for minor revision

---

## Round 1 · Referee Report · Anonymous (Referee 2) · 2024-5-5

Strengths

1) Comprehensive introduction to the method discussing all relevant aspects

2) Discussion of a relevant idea that can improve the well established method of HTSE

3) Completely self-contained article. All relevant proofs and definitions can be found in the appendix.

Weaknesses

1) Many technical details are only discussed in the specialized notation. Examples and more illustrations would help to better communicate the information.

2) The introduction including the Hubbard model is somehow detached from the main body of the article. A focus on the relevance of the HTSE method and some remarkable results obtained using the method would be preferable.

3) There are many technical details that require to jump between the main body of the manuscript and the appendix (and vice versa).

4) There are several small issues regarding the grammar.

5) Regarding the introduction of many mathematical symbols and abbreviations the formatting of the text, equations and definitions is not ideal.

Report

The manuscript describes an idea to improve graph-based high-temperature series expansions (HTSE) for magnetic spin Hamiltonians in a magnetic field.
The main result of the work is a scheme to deduce the contribution for bridged and tree graphs with an improved algorithmic complexity of \(n^2\).
These graphs are required for the HTSE of Heisenberg Hamiltonians in a magnetic field.
With the presented scheme and its improvements the authors claim a benefit of one additional order in the series expansion.

The manuscript is a technical description of a method, therefore there are no "physical examples" discussed.

The manuscript falls under the "Expectations" acceptance criteria 3. and 4. as one can argue that it provides a theoretical/computational advantage to improve existing HTSE by an order.
Since graph-based series expansions are a hard problem this improvement is significant.

Nevertheless, the manuscript still requires some minor changes in order to meet the "general" acceptance criteria of SciPost Physics.

Requested changes

General points:

1) Could you address the points raised in the weaknesses section? These points refer to the entire manuscript.

2) To reach a "bigger audience", it would be very useful to visualize many of the graph-related concepts in figures.

3) It is completely acceptable that there is no big emphasis on physical results in the manuscript. Nevertheless, it would be beneficial to have some connection points to "relevant physics" in the introduction and the summary/outlook.

Specific points:

1) Introduction: Regarding the wording: "we still do not get a solvable model in presence of frustration (competing interactions)". Many non-frustrated models are also not solvable (Heisenberg model on the cubic lattice). Could you please clarify the statement?

2) Introduction: Regarding the sentence: "Frustrated spin models are realized in numerous new materials and exhibit various unconventional phases". Could you please provide some (overview) references to give interested reader an easy access to this statement.

3) Introduction: Regarding the sentence: "Understanding these systems requires increasingly sophisticated methods, including variational methods, mean-field methods, tensor-product numerical methods, and renormalization group methods, among others". Could you please provide some (overview) references to give the interested reader easy access to these methods.

4) Introduction: Regarding the sentence: "Furthermore, extrapolation techniques have been developed to extend the analysis to lower temperatures, necessitating the inclusion of the largest possible number of coefficients in the series". Could you please provide the relevant references.

5) Sec. 2: Regarding the sentence: "2-spin or multispin interactions are possible". Please explain how multispin interactions are incorporated in the graph expansion scheme?

5) Sec. 4: Add a more descriptive caption to Figure 2.

6) Sec. 4: At the end of the paragraph "of the usual method" is mentioned. Please add more description and references to this statement.

7) Appendix C.3: "We now explain a better criterium (C.5), and give an algorithm to compute it.". A better criterium for what? Plase clarify the sentence.

8) Appendix C.3: "Condition “if one of them . . . to an odd islet” is important". Please clarify the condition.

Typos, Style and Grammar:

These are some typos / sentences that caught the eye during the examination of the manuscript. They can be used to improve the manuscript. The authors can adjust their manuscript, but do not need to address these suggestions "point-by-point".

  • General: Often a sentence structure like "We [verb] here ..." is chosen. It would improve the clarity of the sentences if "Here, we [verb] ..." would be used instead.

  • General: Sometimes required articles (a, an, the) are missing in sentences.

  • Title: "algorithm" -> "an algorithm"

  • Introduction: "limited to spin" -> "limited to its spin"

  • Sec. 2.2: "Series expansions of the previous subsection" -> "The series expansions described in the previous section"

  • Sec. 2.2: "is a sum over connected multi-graph" -> "is a sum over connected multi-graphs"

  • Sec. 2.2: "it possesses a well defined HTSE in the thermodynamic limit" -> "it results in a well defined HTSE in the thermodynamic limit"

  • Sec. 2.2: Eq.~(14) -> added "." at the end

  • Sec. 2.2: "To simplify the notations in this presentation" -> "To simplify the notations in this manuscript"

  • Sec. 2.3: "coefficient of" -> "the coefficient of"

  • Sec. 2.3: "are not equal, denominator of coefficient" -> "are not equal, the denominator of the coefficient"

  • Sec. 2.3: "To store in a uniform way the series" -> "To store the series in a uniform way"

  • Sec. 2.3: "and publicly available" -> "and publicly available results"

  • Sec. 2.4: "that do the job" -> "that does the job"

  • Sec. 2.4: "if we keep or not a vertex" -> "if we keep a vertex or not"

  • Sec. 2.4: "It uses the McKay’s algorithm" -> "Canonical labels can be calculated using McKay’s algorithm"

  • Sec. 2.4: "They consist roughly in first generating" -> "They consist roughly of a first step generating"

  • Sec. 2.5: Maybe replace "In the sequel" by "in the following"

  • Sec. 4: The hyperref to Eq.~(34a) and Eq.~(34b) does not work.

  • Appendix A: The listing of definitions might be better as a bulletpoint structure.

  • Appendix B: "but that what follows" -> "but what follows"

  • Appendix B.5: "without isolated site" -> "without an isolated site"

  • Appendix B.5: "Their exponentials too." is not a sentence.

  • Appendix B.6: "Then with Eq.~(B.13) we see that cumulant too." is not a proper sentence.

  • Appendix B.7: "According to induction hypothesis" -> "According to the induction hypothesis"

  • Appendix B.7: "Only remains what we want to prove." is not a proper sentence.

  • Appendix C: There is an unclear structure around Eq.~(C1). Should it be a "," or a "." in the equation?

  • Appendix C.3: "Multigraph U is graph G where some links are doubled". Please clarify the sentence.

  • Appendix D.1: "But there is simpler way" -> "But there is a simpler way"

  • Appendix D.1: The formatting style around the C Code example is not ideal to follow the argument.

  • Appendix D.1: "We can still save half computational time" -> "We can still save half the computational time"

  • Appendix D.2: "since first multiplicand is allways" -> "since the first multiplicand is always"

  • Appendix D.2: "and coefficient" -> "and the coefficient"

Recommendation

Ask for minor revision

---

## Round 1 · Referee Report · Anonymous (Referee 3) · 2024-5-6

Strengths

1) Interesting description of an extension of high-temperature series expansions in the presence of a finite field

Weaknesses

1) Article is very technical 2) Introduction is not very convincing 3) References are poorly present

Report

The article entiteled "High temperature series expansions of S = 1/2 Heisenberg spin models: algorithm to include the magnetic field with optimized complexity" by Pierre, Bernu, and Messio describes algorithmic progress of high-temperature expansions in the presence of a finite magnetic field. The described method is interesting and certainly valuable for publication. It has further been already used by the same authors in previous articles. At the same the present article is purely technical and does not contain any new physical results (as also clearly stated by the authors. In my opinion the article is therefore better suited for SciPost Core.

Requested changes

1) The first paragraph about Hubbard models is completely detached from the rest of the paper. Of course, it is one prominent way to obtain effective spin models, but there are many others. 2) There are almost no references in the introduction, e.g. no link is given to existing literature on (numerical/non-perturbative) linked cluster expansions there are many other methods mentioned without refererence. 3) Page 3, "sec." -> "Sec." 4) Page 3, "bidimensional" -> "two-dimensional" 5) Page 3, "don't" -> "do not" 6) Page 4, "(Note..." -> "(note...)" 7) Page 5 (but also everwhere in the article): check "," and "." after equations, e.g. after (8) and (10) 8) Page 5: "measure, (5) and (6)" -> "measure (5) and (6)" 9) Page 6: "multi-graph U" -> "multi-graphs U" 10) Page 7: there are several methods applying non-perturbative linked-cluster expansions, e.g. check the recent work in SciPost

M. Hörmann, K. P. Schmidt Projective cluster-additive transformation for quantum lattice models SciPost Physics 15, 097 (2023)

and references therein.

11) Page 11: I find the logic a bit strange that one states that (22) is now proven, but then continues with proving (26) which is given half a page later. Maybe one can (26) a bit close to this statement.

12) Page 13: "anti-ferromagnetic" -> "antiferromagnetic"

Recommendation

Accept in alternative Journal (see Report)

---

## Round 1 · Referee Report · Anonymous (Referee 4) · 2024-5-9

Strengths

1) Detailed description of an algorithm for HTSE of high order. An especially interesting development is the inclusion of other important terms apart from the isotropic Heisenberg term in the expansion (Zeeman term, Dzyaloshinskii–Moriya term, and the possibility of obtaining HTSE for Heisenberg XXZ model)

2) An algorithm is open. The Authors present a library of HTSE coefficients for several models on different lattices.

Weaknesses

1) Manuscript is a bit too technical.

2) Some minor grammatical inaccuracies.

Report

As for today, there are two open algorithms for HTSE of S=1/2 Heisenberg models with several nonequivalent interactions [3,12]. Both algorithms allow users to calculate HTSE for \ln Z of 10th order [3] and 12th order for the statical structure factor [12], typically of up to four different exchange interactions. In the submitted manuscript, the Authors improved the order of HTSE over existing ones (for most of the lattices shown in the data repository, improvement is quite significant). A major part of the scientific literature on the HTSE, including many papers by the Authors, was mainly focused on studying the Heisenberg model in zero magnetic field. The presence of a magnetic field in the expansion provides an opportunity to study, for example, the magnetization process. Another interesting result is the possibility of including the Dzyaloshinskii–Moriya term in the expansion (this should be very significant in analyzing the experimental data by the HTSE).

The manuscript is designed as a self-consistent description of the algorithm. Therefore, it is very formal, and sometimes it is hard to follow all the technicalities.

Overall, the manuscript is an important development in the high-temperature studies of spin models, and I support its publication. At the same time, I am asking the Authors to consider improving some points (see below).

Requested changes

1) Title "High temperature series expansions of S = 1/2 Heisenberg spin models: algorithm to include the magnetic field with optimized complexity" --> "High-temperature series expansions of S = 1/2 Heisenberg spin models: an algorithm to include the magnetic field with optimized complexity"

Change high temperature --> high-temperature throughout the manuscript.

2) In Section 2, the Authors mentioned: "... and the interactions are short-range (in practice, first, second, third neighbors)."

Could the Authors be more precise about how many different exchange interactions are possible to get HTSE of a reasonably high order? 

3) In Section 2, the Authors stated: "Nevertheless, B is an experimentally adjustable parameter that has been known to induce various unexpected phenomena such as magnetization plateaus and phase transitions."

Could the Authors elaborate on this statement? If the order of expansion is sufficiently high to get to rather low temperatures (let's say T=0.2J), could one see "melted" by temperature magnetization plateaus?

4) In Appendix 3, in equation (B.3) limits of the sum are missing \sum_{j=1}^{k}.

Recommendation

Ask for minor revision

---

## Round 2 · Author Response

Warnings issued while processing user-supplied markup:
- Inconsistency: Markdown and reStructuredText syntaxes are mixed. Markdown will be used.
Add "#coerce:reST" or "#coerce:plain" as the first line of your text to force reStructuredText or no markup.
You may also contact the helpdesk if the formatting is incorrect and you are unable to edit your text.
Following the referees comments, we have largely rewritten the introduction and some sections with dense mathematical content. We have tried to answer at best to the relevant comments of the four referees. We feel our article is now clearer both for readers who discover high temperature series expansions and want to have an insight into the algorithm that calculates it, and for the advanced readers who may want to know the details of mathematical assertions. Below are the answers to each referee.
Answer to report 1
We thank the referee for the very positive appreciation of our article. We have taken into account the referee comments:
1) introduction is too short and technical - We agree that the introduction was not adapted to the content of the article. We have fully rewritten it and added references on the milestones of the HTSE. We give more details on the way to use the series obtained by our algorithm, with references to articles where they have already been used. 2) no actual results are presented - We have exploited the series obtained with our algorithm on some specific models (3d Heisenberg models, kagome models) in previous publications. To present actual results in this article, we give the linear magnetic susceptibility from series on three lattices, for several values of the magnetic field. Only the partial sum and the Pade approximants are presented, as the use of extrapolation methods such as the entropy method is out of the reach of the present article, that focus on the obtention of the series. We also more clearly say that high order series are made available by us, at precedently unknown orders (see new Table 2). We feel that the exploitation of these series is not the subject of this article and let it for future use, eventually with experimental relevance.
Answer to report 2
We thank the referee for the very careful reading of our work and for raising important points which helped us to improve our manuscript. We answer below each point raised (W means a point in the Weaknesses section), and have taken into account all the typos, style and grammar remarks.
"W1) Many technical details are only discussed in the specialized notation. Examples and more illustrations would help to better communicate the information." "2) To reach a "bigger audience", it would be very useful to visualize many of the graph-related concepts in figures." "W5) Regarding the introduction of many mathematical symbols and abbreviations the formatting of the text, equations and definitions is not ideal."
These three remarks refer to the complexity for the reader to keep in mind all the notations used, related to the difficulty to apprehend them on simple examples. We have suppressed some notations that were not essential, thus reducing their number (for example, the $\tidle J$ and $\tilde h$). The figure illustrating some graph-related definitions has been integrated in the main text (it was previously in the appendix), and and two new ones have been added (fig. 4 to illustrate Eq. 17 on a simple example, and another one to illustrate Eq. 18). Moreover, we have fully reformulated Sec. 2.3 and structured independent sub-parts into proof, remarks and paragraphs (and we did the same for App.C.3).
1) Introduction: Regarding the wording: "we still do not get a solvable model in presence of frustration (competing interactions)". Many non-frustrated models are also not solvable (Heisenberg model on the cubic lattice). Could you please clarify the statement?
We have replaced the sentence "However, even for spin interactions as simple as the Heisenberg ones, we still do not get a solvable model in presence of frustration (competing interactions)" by "However, even for spin interactions as simple as Heisenberg, the nature of the ground state is still debated on some non bipartite antiferromagnetic lattices. In these cases, frustration prevents the use of exact methods (exact means here with only statistical errors) such as path integral quantum Monte Carlo or stochastic series expansions"
"W2) The introduction including the Hubbard model is somehow detached from the main body of the article. A focus on the relevance of the HTSE method and some remarkable results obtained using the method would be preferable."
We agree that the introduction was not adapted to the content of the article and have purely removed the first sentence about the Hubbard model. We have fully rewritten the introduction and added references on the milestones of the HTSE. We give more details on the way to use the series obtained by our algorithm (using for example the entropy extrapolation method), with references to articles where they have been used.
"W3) There are many technical details that require to jump between the main body of the manuscript and the appendix (and vice versa)."
We have reread the article with this remark in mind and have adapted the sentences with references to the appendices. We now state precisely when it is the proof of a statement, or when it is related to a general property of the cumulants that we explain. We feel that is is now possible to read the main article without jumping to the appendices.
"3) It is completely acceptable that there is no big emphasis on physical results in the manuscript. Nevertheless, it would be beneficial to have some connection points to "relevant physics" in the introduction and the summary/outlook. "
The authors and other researchers have exploited HTSE on some specific models (3d Heisenberg models, kagome models) in previous works, whose citations are now more emphasized. In many cases, it was in view of exploiting experimental results on compounds. To present actual results in this article (for an illustration purpose), we give the linear magnetic susceptibility from series on three lattices, for several values of the magnetic field. Moreover, we give more details in the conclusion about the way series have been used in the presence of high magnetic fields to fit the specific heat on the kagome antiferromagnet Herbertsmithite.
"4) Introduction: Regarding the sentence: "Furthermore, extrapolation techniques have been developed to extend the analysis to lower temperatures, necessitating the inclusion of the largest possible number of coefficients in the series". Could you please provide the relevant references."
This is related to our answer to remark W2. The extrapolation methods are now more detailed and referenced, and the important effect of adding even one or two order is emphasized: "We insist on the fact that knowing a series up to some order n means that correlations in any size-n cluster is exactly taken into account. Calculating one more coefficient is hard, but brings a real constraint on the extrapolations. "
"5) Sec. 2: Regarding the sentence: "2-spin or multispin interactions are possible". Please explain how multispin interactions are incorporated in the graph expansion scheme?"
We have replaced this elusive sentence on multispin interactions by " HTSE can in principle be calculated for any type of 2-spin interaction (with anisotropies, Dzyaloshinskii-Moriya... ), even if only Heisenberg interactions are considered in the following. Multispin interactions (also called ring or cyclic exchange) are possible, but their presence would considerably increase the complexity of the step where graphs are enumerated. In this case, at order n in β, graphs would not only be constituted of n elementary block of site or link type, but also of plaquette type (of typically 4 of 6 links). "
"5) Sec. 4: Add a more descriptive caption to Figure 2."
Caption of figure 2 (now Fig. 5) has been consequently expanded.
"6) Sec. 4: At the end of the paragraph "of the usual method" is mentioned. Please add more description and references to this statement."
The imprecise words "usual method" has been replaced by "method used for any graph in Sec. 2.5."
"7) Appendix C.3: "We now explain a better criterium (C.5), and give an algorithm to compute it.". A better criterium for what? Plase clarify the sentence."
We have replaced "a better criterium (C.5)" by "a criterion (C.4) that allows to discard more graphs than (C.3)" and accordingly modified the title of the subsection C.3. Moreover, we have clarified the structure of this subsection by writing the criterion at the beginning and then clearly separating its proof and the algorithm to apply it.
"8) Appendix C.3: "Condition “if one of them . . . to an odd islet” is important". Please clarify the condition."
We have largely clarified section C3 and give more details on the algorithm to find the optimal multigraph U.
Answer to report 3
We thank the referee for the very careful reading of our work and for raising important points which helped us to improve our manuscript. We answer below each point raised (W means a point in the Weaknesses section). We have taken into account all the typos, style and grammar remarks.
"W1) Article is very technical W2) Introduction is not very convincing W3) References are poorly present" 2) There are almost no references in the introduction, e.g. no link is given to existing literature on (numerical/non-perturbative) linked cluster expansions there are many other methods mentioned without refererence."
We have fully rewritten the introduction to make the technicality of our article more meaningful and to put in context the usefulness of obtaining numerous coefficients from the series. In this new introduction, we refer to many articles that have calculated or used series in a way to advance knowledge on magnetic systems. To still smooth the technical aspect, we now present direct applications of the coefficients obtained with our algorithm by presenting curves of the linear magnetic susceptibility on three lattices, for several values of the magnetic field.
"1) The first paragraph about Hubbard models is completely detached from the rest of the paper. Of course, it is one prominent way to obtain effective spin models, but there are many others."
We have removed this paragraph.
"3) Page 3, "sec." -> "Sec." 4) Page 3, "bidimensional" -> "two-dimensional" 5) Page 3, "don't" -> "do not" 6) Page 4, "(Note..." -> "(note...)" 7) Page 5 (but also everwhere in the article): check "," and "." after equations, e.g. after (8) and (10) 8) Page 5: "measure, (5) and (6)" -> "measure (5) and (6)" 9) Page 6: "multi-graph U" -> "multi-graphs U" 12) Page 13: "anti-ferromagnetic" -> "antiferromagnetic"
We have corrected all these typos and thanks the referee for having listed them.
"10) Page 7: there are several methods applying non-perturbative linked-cluster expansions, e.g. check the recent work in SciPost M. Hörmann, K. P. Schmidt Projective cluster-additive transformation for quantum lattice models SciPost Physics 15, 097 (2023) and references therein."
The only cited example of linked-cluster expansions (the numerical linked-cluster expansion) has been chosen due to its similarity with the HTSE. However, we agree that other valuable methods exists, and have added the example proposed by the referee.
"11) Page 11: I find the logic a bit strange that one states that (22) is now proven, but then continues with proving (26) which is given half a page later. Maybe one can (26) a bit close to this statement."
We have removed the reference to Eq. (26) (now number (24)). This equation is now just a part of the 'proof' paragraph for Eq. (22) (now number (20)).
Answer to report 4
We thank the referee for the very careful reading of our work and for raising important points which helped us improve our manuscript. We answer below each point raised (W means a point in the Weaknesses section).
W1) Manuscript is a bit too technical.
This is a comment common to all referees, that has led us to: - fully rewrite the introduction to make the technicality of our article more meaningful - illustrate our results by presenting direct applications of our algorithm: we present HTSE curves of the linear magnetic susceptibility on three lattices, for several values of the magnetic field. - suppress some notations that were not essential, thus reducing their number (for example, the $\tidle J$ and $\tilde h$). The figure illustrating some graph-related definitions has been integrated in the main text (it was previously in the appendix), and two new ones have been added (fig. 4 to illustrate Eq. 17 on a simple example, and another one to illustrate Eq. 18).
W2) Some minor grammatical inaccuracies. 1) Title "High temperature series expansions of S = 1/2 Heisenberg spin models: algorithm to include the magnetic field with optimized complexity" --> "High-temperature series expansions of S = 1/2 Heisenberg spin models: an algorithm to include the magnetic field with optimized complexity" Change high temperature --> high-temperature throughout the manuscript.
We have done these modifications and tried to correct the grammatical inaccuracies.
2) In Section 2, the Authors mentioned: "... and the interactions are short-range (in practice, first, second, third neighbors)." Could the Authors be more precise about how many different exchange interactions are possible to get HTSE of a reasonably high order?
We have removed "in practice" in this introductive sentence about models that are possible in principle, as the number of neighbors "only" impacts the computation time: HTSE works in principle for any (short) range. The impact on the computation time is discussed two paragraphs later: "The computation time depends on the model: lattice geometry, spin length and interaction type. The coordination number of each site (related to the lattice geometry and the type of links: first, second... neighbors) determines the evolution of the graph number with the order,"
To answer the question of the referee, it is difficult to be more precise about this point, as the order highly depends on the connectivity of the lattice. Considering more types of exchanges considerably increases the number of coefficients in the polynomials (terms of the series are polynomials in J1, J2, J3, J4...) and accordingly, increases the time of the trace calculation. We have for example obtained the series on kagome for J1, J2, J3 and J3h (two types of third neighbor interactions) up to order 8. But on other lattices, 4 exchanges are generally too much. In the HTSE coefficients publicly available, one can find those of many models with 2nd or 3rd neighbors, and see the orders that have been obtained.
3) In Section 2, the Authors stated: "Nevertheless, B is an experimentally adjustable parameter that has been known to induce various unexpected phenomena such as magnetization plateaus and phase transitions." Could the Authors elaborate on this statement? If the order of expansion is sufficiently high to get to rather low temperatures (let's say T=0.2J), could one see "melted" by temperature magnetization plateaus?
This article is the first step to study spin models using HTSE when h varies: the computation of series for any h. In a second step, we plan to apply and ameliorate extrapolation technics (namely the entropy method) to study in details some models in the (T, h) plane (kagome, square kagome, as in https://journals.jps.jp/doi/abs/10.7566/JPSJ.91.094711 that we now cite). Seeing melted magnetization plateaus would effectively be a very promising result, but is out of reach of the present article, as can be understood from Fig.1, and is explained in the added sentence: "This highlights the importance of in a first step computing HTSE with non-zero magnetic fields, which is the subject of this article and in a second step, developing powerful HTSE extrapolation technics to explore the (T,h) plane. Fig.1 emphasizes the necessity of such extrapolations: temperatures accessible by naive extrapolation techniques decrease with $h$ and do not allow to reach temperatures where peaks in $\chi_l(h)$ appear, precursors of $T=0$ plateaus."
4) In Appendix 3, in equation (B.3) limits of the sum are missing \sum_{j=1}^{k}.
This has been corrected.

---

## Round 2 · List of Changes

- The introduction was not adapted to the content of the article and this was a remark from several referees. We have fully rewritten it and added references on the milestones of the HTSE. We give more details on the way to use the series obtained by our algorithm, with references to articles where they have already been used.
- A new figure presents the linear magnetic susceptibility from series on three lattices, for several values of the magnetic field, to give an application of our series coefficients.
- To lighten the mathematical complexity, we have suppressed some notations that were not essential, thus reducing their number (for example, the $\tidle J$ and $\tilde h$). The figure illustrating some graph-related definitions has been integrated in the main text (it was previously in the appendix), and a two new ones have been added (fig. 4 to illustrate Eq. 17 on a simple example, and another one to illustrate Eq. 18). Moreover, we have fully reformulated Sec. 2.3 and structured independent sub-parts into proof, remarks and paragraphs and similarly for Sec. C.3.
- A new table (Tab. 2) gives the orders that we have reached with our algorithm for some models with Heisenberg first neighbor interactions.
- The grammatical and lexical corrections of the referees have been taken into account.

---

## Editorial Decision

published